# Direct observation of tRNA-chaperoned folding of a dynamic mRNA ensemble

Krishna C. Suddala [1,6], Janghyun Yoo [2,6], Lixin Fan[3], Xiaobing Zuo [4], Yun-Xing Wang [3,5], Hoi Sung Chung [2] ✉ & Jinwei Zhang [1] ✉

T-box riboswitches are multi-domain noncoding RNAs that surveil individual amino acid availabilities in most Gram-positive bacteria. T-boxes directly bind specific tRNAs, query their aminoacylation status to detect starvation, and feedback control the transcription or translation of downstream amino-acid metabolic genes. Most T-boxes rapidly recruit their cognate tRNA ligands through an intricate three-way stem I-stem II-tRNA interaction, whose establishment is not understood. Using single-molecule FRET, SAXS, and time-resolved fluorescence, we find that the free T-box RNA assumes a broad distribution of open, semi-open, and closed conformations that only slowly interconvert. tRNA directly binds all three conformers with distinct kinetics, triggers nearly instantaneous collapses of the open conformations, and returns the T-box RNA to their pre-binding conformations upon dissociation. This scissors-like dynamic behavior is enabled by a hinge-like pseudoknot domain which poises the T-box for rapid tRNA-induced domain closure. This study reveals tRNA-chaperoned folding of flexible, multi-domain mRNAs through a Venus flytrap-like mechanism.

T-box riboswitches are *cis*-acting, gene-regulatory non-coding RNAs present in most Gram-positive bacteria including many clinically important human pathogens[1–5]. T-boxes directly bind to specific tRNAs, sense their aminoacylation status, and regulate the transcription or translation of downstream genes involved in amino acid biosynthesis, transport, and tRNA aminoacylation[6]. Most T-boxes contain three principal domains: a 5′ stem I domain, an intervening stem II domain, and a 3′ discriminator domain. Stem I invariably contains an mRNA codon-like "Specifier" trinucleotide sequence complementary to the tRNA anticodon. This Specifier-anticodon base-pairing interaction is mainly responsible for the specific selection of cognate tRNAs. Binding is augmented by either a stem I-distal interdigitated double T-loop motif (IDTM) that stacks with the tRNA elbow[7–9], or a stem II S-turn motif that bridges and stabilizes the Specifier-anticodon pairing[10], or both. The stem II domain is connected to stem I via a

conserved, sharply bent Kink turn (K-turn) and contains a long stem (stem II) immediately followed by a pseudoknot (stem IIA/B, Ψ) that stacks coaxially with stem II[10]. The 3′ discriminator domain, consisting of the stem III and an anti-terminator or anti-sequestrator module, captures the tRNA 3′ end, senses its aminoacylation status, and switches conformations to regulate transcription termination or translation initiation of downstream genes, respectively[11].

There are at least four extant classes of T-boxes, classified based on their divergent stem I–stem II architectures (Fig. 1a)[6,7]. While Class I and II T-boxes generally regulate transcription termination, Class III and IV T-boxes mostly regulate translation initiation. Class I T-boxes, exemplified by the first discovered *Bacillus subtilis tyrS* T-box, feature a long stem I harboring a distal IDTM, host the Specifier in an internal loop, and contain a stem II domain bearing an S-turn. Class II T-boxes, represented by the *B. subtilis glyQS* T-box, resemble Class I except that

[1]Laboratory of Molecular Biology, National Institute of Diabetes and Digestive and Kidney Diseases, Bethesda, MD 20892, USA. [2]Laboratory of Chemical Physics, National Institute of Diabetes and Digestive and Kidney Diseases, Bethesda, MD 20892, USA. [3]Basic Science Program, Frederick National Laboratory for Cancer Research, Small-Angle X-Ray Scattering Core Facility of National Cancer Institute, Frederick, MD 21702, USA. [4]X-ray Science Division, Argonne National Laboratory, Lemont, IL 60439, USA. [5]Structural Biophysics Laboratory, Center for Cancer Research, National Cancer Institute, Frederick, MD 21702, USA. [6]These authors contributed equally: Krishna C. Suddala, Janghyun Yoo ✉e-mail: chunghoi@niddk.nih.gov; jinwei.zhang@nih.gov

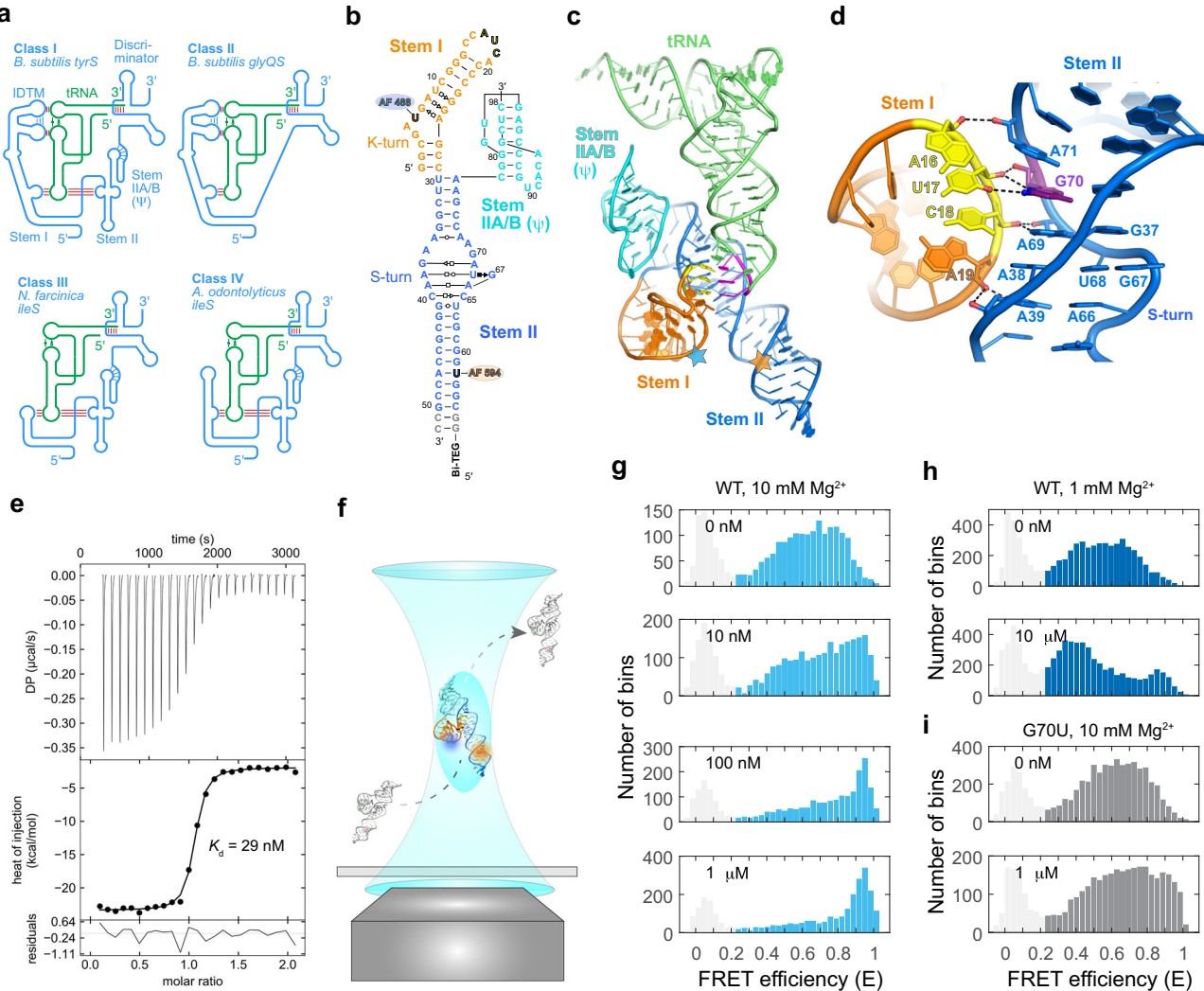

**Fig. 1 | smFRET analysis of solution conformations of the *ileS* T-box riboswitch.** **a** Four classes of T-box riboswitches with different structural features. IDTM: interdigitated double T-loop motif. T-box-tRNA contacts are indicated by red sticks. **b** Sequence and secondary structure of the two-piece *Nocardia farcinica ileS* T-box riboswitch construct used in smFRET analyses. Fluorophore labeling positions are indicated. **c** Co-crystal structure of the *N. farcinica ileS* T-box riboswitch in complex with tRNA[Ile] (PDB: 6UFM). **d** Rotated, detailed view of the stems I–II docking interface showing the ribose zipper interactions. tRNA is omitted for clarity. Hydrogen bonds are indicated by dashed lines. **e** Isothermal titration

calorimetry (ITC) measurement of tRNA[Ile] binding to the WT two-piece T-box riboswitch construct. **f** Schematic of free-diffusion confocal smFRET analysis. **g–i** FRET efficiency histograms of **g** WT T-box RNA with 10 mM Mg$^{2+}$ in the absence or presence of 10 nM, 100 nM, or 1 μM tRNA, **h** WT T-box RNA with 1 mM Mg$^{2+}$ in the absence or presence of 10 μM tRNA, and **i** G70U T-box RNA with 10 mM Mg$^{2+}$ in the absence or presence of 1 μM tRNA. The histograms were constructed using bins (2 ms bin time) containing 30 photons or more. The peak at $E \sim 0.1$ (light gray) in the histograms results from the molecules lacking the acceptor. FRET efficiencies were corrected for background. Source data are provided as a Source Data file.

they are glycine-specific and substitute a flexible single-stranded (ss) linker for the stem II domain. Class III T-boxes, represented by the *Nocardia farcinica ileS* T-box, have short stem I domains without an IDTM, host the Specifier in a pentanucleotide or hexanucleotide apical loop, and require stem II for function. Class IV T-boxes, such as the *Actinomyces odontolyticus ileS* T-box[12], are similar to Class III except that they feature a stem I of intermediate length, which does not contain an IDTM, and house the Specifier in a side bulge at the junction between two helices of stem I (Fig. 1a).

Two co-crystal structures of Class III T-boxes bound to tRNA[Ile] revealed a surprising trilateral mode of tRNA binding where stems I and II form a tight groove that interacts with the tRNA anticodon stem–loop (ASL) (Fig. 1b, c)[10,13]. Despite the absence of the IDTM, which contacts tRNA elbow in Class I and II T-boxes and confers binding avidity, this localized tRNA interaction achieves ~6-fold higher affinity ($K_d \sim 24$ nM)[7]. In the crystal structure of *N. farcinica ileS* T-box, stem I and stem II intimately dock with each other via an extended ribose-

zipper interaction (Fig. 1d), which features multiple hydrogen bonds between the stem I Specifier trinucleotide and stem II S-turn region involving their ribose 2′-OH groups, sugar edges of nucleobases, and phosphate oxygens.

However, it is unknown how this high-affinity, three-way interaction among the stem I, stem II, and tRNA is established. At least three potential scenarios can occur. It is possible that stems I and II are stably docked forming a largely static, compact structure, which recruits the tRNA using a pre-organized binding groove. Alternatively, stems I and II can be in a dynamic, rapid docking and undocking equilibrium, and only the docked conformer can bind tRNA ASL and shift the equilibrium towards binding—a conformational selection model. In a third scenario, the tRNA may be able to engage stem I or II first in open T-box conformations and subsequently induce the docking of the two stems to prevent its own dissociation—an induced fit mechanism. To distinguish among these potential scenarios, and gain a physical picture of how structured, multi-domain RNAs dynamically establish multilateral,

multivalent contacts, we employed two- and three-color single-molecule Förster resonance energy transfer (smFRET) spectroscopy, small-angle X-ray scattering (SAXS) and time-resolved 2-aminopurine (2AP) fluorescence spectroscopy analyses to probe the global and local conformations and dynamics of an *N. farcinica ileS* T-box riboswitch during tRNA binding and dissociation. We find that the free T-boxes exist in a broad distribution of open, semi-open, and closed conformers, all of which can bind tRNAs but with distinct kinetics and tend to retain the memory of their pre-binding conformations. tRNA binding triggers rapid closure of open T-box conformations that are orchestrated by a region composed of the stems I–II junction and stem IIA/B pseudoknot, which serves as a hinge and backstop to enable the scissors-like motions of the T-box mRNA. Together, this study provides mechanistic insights into how multivalent RNA-RNA interactions are dynamically established to regulate gene expression during nutritional stress and provides a proof-of-concept that structured RNAs can guide the folding of complex, multi-domain noncoding RNAs.

## Results

### Free-diffusion smFRET reveals multiple conformations of the *ileS* T-box riboswitch

Structured RNA molecules are frequently conformationally fluidic and can adopt multiple conformations that populate a rugged free-energy folding landscape[14,15]. smFRET analyses have been widely used to characterize the conformational ensemble, structural dynamics, and ligand-mediated folding of diverse non-coding RNAs, including metabolite-binding riboswitches and transcription-regulating Class II T-boxes[16–19]. To probe the solution conformation of Class III *N. farcinica ileS* T-box riboswitch by smFRET, we designed a two-piece construct internally labeled with donor (Alexa Fluor 488) and acceptor (Alexa Fluor 594) fluorophores in stems I and II, respectively (Fig. 1b, c). ITC measurements using the two-piece T-box design reported a $K_d$ value of 29 nM (Fig. 1e), nearly identical to the 24 nM affinity reported for the wild-type (WT) one-piece T-box, suggesting that the two-piece design did not perturb tRNA interactions[10].

To delineate the equilibrium distribution of the free T-box conformations in solution, we first performed smFRET measurements on freely diffusing molecules (Fig. 1f). In the absence of tRNA but in the presence of 10 mM $Mg^{2+}$, the FRET efficiency (*E*) histogram for the apo T-box riboswitch showed a broad distribution of different FRET efficiency values. (Fig. 1g). Addition of tRNA led to a dose-dependent increase in the population of the state with a high-FRET efficiency (*E* > 0.85) with a concomitant decrease in the fraction of lower-FRET conformations. In the presence of 1 µM tRNA (~41 fold above $K_d$), ~90% of T-box molecules adopt the high-FRET conformation. The high-FRET value of 0.95 corresponds to a short distance of <35 Å between the fluorophores and likely represents a compact, closed conformation as seen in the co-crystal structures where stems I and II are docked via the ribose zipper, with the distance between the labeling sites being ~28 Å (Fig. 1c, d)[10,13]. Notably, the FRET analysis only reports on the global conformation and does not possess the local information to distinguish between the truly docked (interacting) conformation and another globally closed but not docked conformation that may also exist. Therefore, the high-FRET state is "closed" but not necessarily docked. By contrast, the heterogeneous, lower FRET states presumably represent 'open' and 'semi-open' conformations of T-box, with larger separation between the stems (see the immobilization experiment section below for the definition of these states). At a lower $Mg^{2+}$ concentration of 1 mM, the FRET distribution of the apo T-box riboswitch was similarly broad but shifted towards lower FRET states, showing that the T-box adopts more extended conformations under reduced $Mg^{2+}$ (Fig. 1h). Under these conditions, the addition of tRNA, as high as 10 µM, only induced a minor shift towards high-FRET states. This finding suggests that tRNA binding to the *ileS* T-box requires higher $Mg^{2+}$ concentrations (>10 mM) in vitro, as previously observed

for this and other T-boxes such as *B. subtilis glyQS*[7,19]. By contrast, the intracellular concentrations of free $Mg^{2+}$ are estimated to be ~1–2 mM in bacteria[20]. This empirically observed, elevated $Mg^{2+}$ requirement in vitro seems to be a general characteristic of complex RNA-RNA and RNA-ligand interactions. A comparative in-cell and in vitro SHAPE (Selective 2′ Hydroxyl Acylation analyzed by Primer Extension) analysis of the ligand-free adenine riboswitch revealed that its in-cell and in vitro conformations correlate poorly at 1 mM $Mg^{2+}$ (*R* = 0.55) but are substantially similar at 5 mM or higher $Mg^{2+}$ (*R* = 0.75)[20]. Therefore, the in-cell environment is uniquely able to promote the folding, structuration, and interactions of RNAs and obviates an otherwise much higher $Mg^{2+}$ requirement. This effect could be mediated by molecular crowding[21], osmolytes such as TMAO (Trimethylamine N-oxide)[22], or RNA-binding proteins such as YbxF, which stabilizes K-turns found in T-boxes[23].

Next, to assess how much stems I–II docking may contribute to the T-box conformation distribution, we disrupted the docking by introducing a G70U substitution in stem II. The conserved G70 sits immediately adjacent to the S-turn near the center of a crescent-shaped 5-purine string that latches the Specifier-anticodon duplex in its minor groove[6,10]. G70 makes three hydrogen bonds to U17 in stem I, two of which involve the sugar edge of its guanosine base (Fig. 1d). Consequently, a G70U substitution reduced tRNA binding by ~135 fold by disrupting the ribose zipper. Remarkably, the G70U apo T-box, which cannot stably dock, exhibited a FRET distribution nearly indistinguishable from the WT T-box (Fig. 1i). This observation suggests that stems I–II docking contributes little to its conformational distribution in the absence of tRNA. As expected, the G70U T-box only slightly shifted towards the closed conformation even with 1 µM tRNA, consistent with its severely compromised ability to bind tRNA (Fig. 1i).

### SAXS analyses reveal extended apo T-box conformations in solution

To further examine the conformations of the T-box in solution using an orthogonal approach, we employed small-angle X-ray scattering (SAXS) analyses of the apo T-box, tRNA, and the T-box-tRNA complex (Fig. 2a, b)[24,25]. Kratky plots for the apo T-box, tRNA, and the complex indicated that these RNA samples are well structured (Supplementary Fig. 1). The radii of gyration ($R_g$) of the apo T-box, tRNA, and T-box-tRNA complex obtained from Guinier analysis were 29.9 Å, 23.4 Å, and 33.4 Å, respectively. To compare in solution and *in crystallo* conformations of the T-box RNAs, we used CRYSOL analyses (Fig. 2a, b). The theoretical scattering profile for the T-box-tRNA complexes calculated by CRYSOL using the crystal structure matched reasonably well with the experimental scattering curve with a lower $\chi^2$ value of ~2.8, suggesting only minor differences exist between the solution conformation and the crystal structure. By contrast, the CRYSOL-predicted scattering profile using the 'docked' conformation of the T-box extracted from the complex structure deviates substantially from the experimental data for the apo T-box, giving a high $\chi^2$ value of ~35.6. This finding suggests that the apo T-box assumes a global conformation in solution distinct from the compact, crystallographically observed conformation with docked stems I and II. Further, the pairwise distance distribution function (PDDF) for the apo T-box indicates a maximum particle dimension ($D_{max}$) of >120 Å, nearly 30 Å longer than the ~92 Å distance estimated from the crystal structure (Fig. 2c, Supplementary Fig. 2). By contrast, the $D_{max}$ of the T-box–tRNA complex derived from PDDF analysis was ~116 Å, which matches closely with the 119 Å longest dimension calculated from the crystal structure. Together, these data suggest that the apo T-box assumes extended conformations in solution that are drastically different from the docked conformation, in agreement with the increased stems I–II separations observed by smFRET. Furthermore, PDDF analysis of the C6U T-box revealed a distance distribution nearly identical to the WT T-box (Supplementary Fig. 3), which affirms that this substitution in

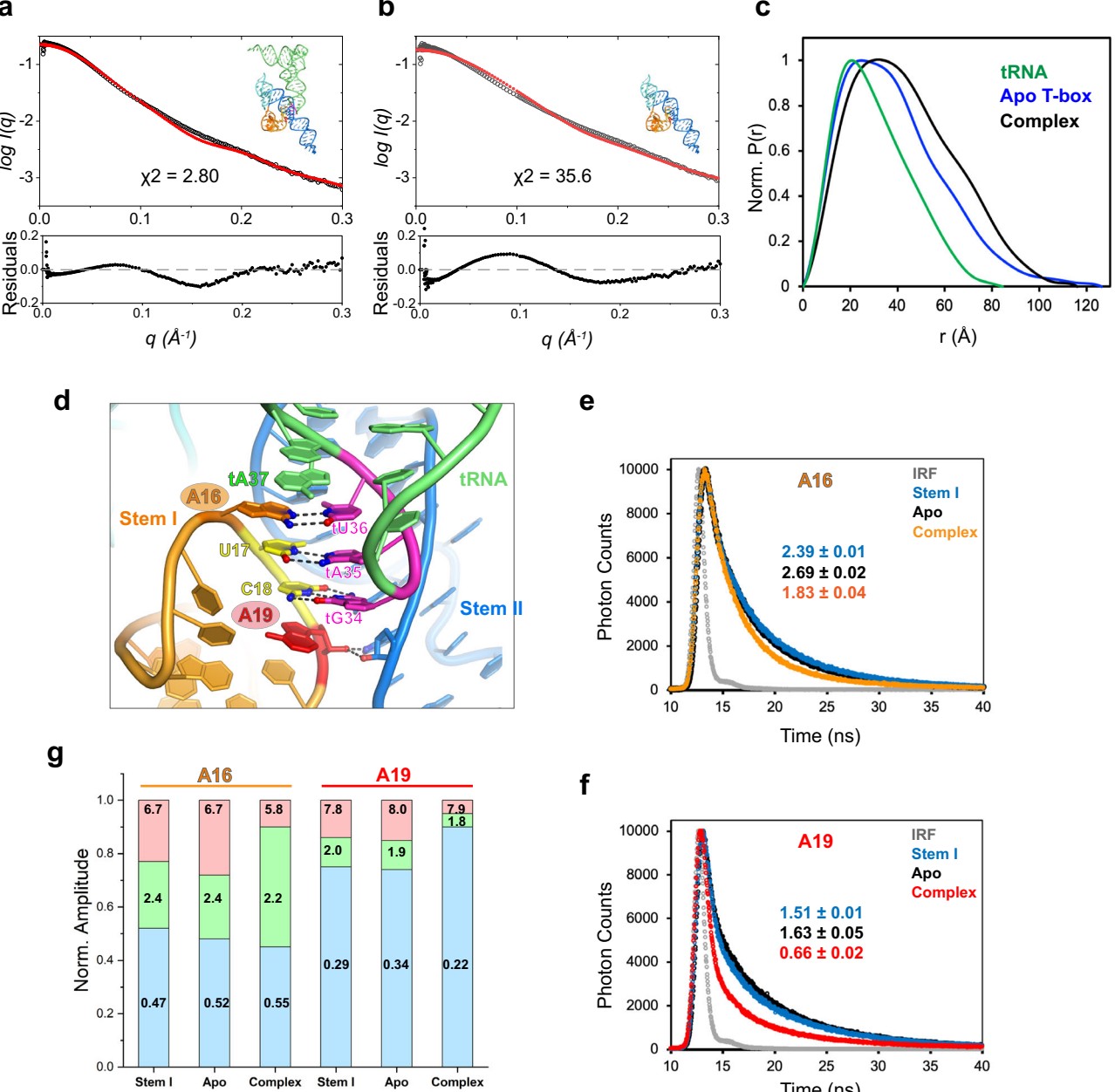

**Fig. 2 | SAXS analyses and 2-aminopurine (2AP) lifetime probing of the T-box RNA. a, b** Overlay of back-calculated SAXS scattering curves (red dotted lines) computed from the cocrystal structure of the *N. farcinica ileS* T-box riboswitch in complex with tRNA^Ile (**a**) or the T-box portion extracted from the complex structure (**b**) using CRYSOL, with experimental SAXS profiles (black dotted lines). Residuals and $\chi^2$ of the fits are indicated. **c** Pair-wise distance distribution function (PDDF) plots for the tRNA (green), apo T-box (blue), and T-box-tRNA complex (black).

**d** Location of the 2AP probes (A16, orange; A19, red) in the co-crystal structure at the T-box Specifier-tRNA anticodon interface. **e, f** Time-resolved fluorescence decay time traces for 2AP at positions A16 (**e**) or A19 (**f**). Amplitude-weighted averaged lifetimes $\tau_{avg}$ are indicated. IRF: instrument response function. **g** Individual lifetimes and relative amplitudes derived from (**e, f**). The amplitudes for the short, intermediate, and long lifetimes are shown in light blue, green, and red, respectively. Source data are provided as a Source Data file.

the stem I K-turn to facilitate fluorophore labeling did not alter the T-box conformation. By contrast, G70U substitution led to a slightly extended conformation, presumably due to the inability of stems I and II to dock, in line with the smFRET findings (Fig. 1i).

## 2-aminopurine lifetime analyses suggest the absence of stems I–II docking in the apo T-box

While the smFRET and SAXS analyses provided congruent information on the T-box global conformations, we also sought to obtain local conformational information near the stems I–II docking interface. 2-aminopurine (2AP) is a fluorescent adenine analog widely used to

probe the local environments and conformations of diverse nucleic acid structures and interactions, including the *B. subtilis glyQS* T-box and other riboswitches[26–29]. The quantum yield and lifetime of 2AP are exquisitely sensitive to stacking interactions with neighboring nucleobases[30]. Compared to intensity-based measurements, time-resolved fluorescence decay analyses are concentration-independent, less prone to optical variances and external interferences, and possess the precision and sensitivity to distinguish between one-sided stacking and two-sided stacking interactions with 2AP[27,31].

We employed the same two-piece T-box design used in the smFRET analyses and substituted adenosines A16 or A19 in the

Specifier loop with 2AP (Fig. 2d). The 2AP substitutions slightly reduced tRNA binding affinity as they alter the T-box base-pairing interface with the tRNA. However, neither 2AP substitution is expected to affect stems I–II docking, which is mediated by the ribose zipper and does not involve the nucleobases of A16 or A19. The decay of 2AP fluorescence in aqueous solutions typically consists of three to four individual exponents[27]. In most cases, three exponents are sufficient to deconvolute the decay curves producing $\chi^2 < 1.3$ (Supplementary Table 2). Subsequently, an amplitude-weighted mean lifetime, $\tau_{avg}$, is computed and reports the extent of 2AP quenching or stacking. The crystal structure indicates that stems I–II docking via the extended ribose zipper coincides with a near helical, stacked conformation of the A16–U17–C18–A19 tetranucleotide in the Specifier loop (Fig. 2d)[10]. Further, the ribose zipper is expected to restrict the motion of the tetranucleotide, enhance intra-strand stacking, and thus reduce the lifetime of 2AP incorporated in this region. Indeed, stems I–II docking in the T-box-tRNA complex led to substantial reductions in the average lifetimes ($\tau_{avg}$) of both 2AP16 (from 2.7 to 1.8 ns, Fig. 2e, g) and 2AP19 (from 1.6 to 0.66 ns, Fig. 2f, g). Then, we measured the 2AP lifetimes in the stem I-only constructs and found that they were nearly the same (2AP16: 2.4 ns; 2AP19: 1.5 ns) as those of the apo T-box (2AP16: 2.7 ns; 2AP19: 1.6 ns). Thus, the addition of stem II did not reduce but slightly increased the lifetimes of both 2APs in stem I, suggesting that stem II, when present, is not stably docked with stem I in the apo T-box. These 2AP lifetime data provide complementary local information consistent with the smFRET and SAXS findings on global conformations.

## Surface-immobilized smFRET visualizes T-box conformer interconversions

While our free-diffusion smFRET experiments described the distribution of multiple T-box conformations and tRNA-induced shifts to high-FRET states, they do not provide kinetic information about potential state interconversions. To visualize the structural dynamics of apo T-boxes, we immobilized them on NeutrAvidin-coated slides and followed them for 5–10 min before photobleaching (Fig. 3a, Supplementary Fig. 4). Interestingly, most of the smFRET time traces for individual molecules showed constant FRET efficiency values with rare transitions to other FRET states (Fig. 3b, Supplementary Fig. 4a). This shows that the different conformational states of the apo T-box do not readily interconvert or do so with slow rates. The FRET efficiency histogram corrected for the background, the donor leak into the acceptor channel, and the quantum yields and detection efficiencies of the donor and acceptor ($\gamma$-factor) resemble those from free-diffusion experiments and showed three distinct states, whose FRET efficiencies and fractional population were further assigned using the two-dimensional FRET efficiency-donor lifetime analysis (Fig. 3c, e, Methods). The broad distribution can be clustered into three conformational states. The low ($E \sim 0.34$, 42 %), mid ($E \sim 0.61$, 48 %), and high ($E \sim 0.91$, 10 %) FRET states correspond to open, semi-open, and closed T-box conformations and have approximate donor-acceptor distances of ~60, 50, and 37 Å, respectively (Methods). The estimated 37 Å distance is consistent with the 28 Å distance between the fluorophore-labeling sites (C5 atoms of C6 in stem I and C58 in stem II) observed in the crystal structure, considering the length of the linker to the fluorophores.

Interestingly, tRNA addition mobilized the T-box conformers (Fig. 3d, Supplementary Fig. 4b). In the presence of 50 nM tRNA, most smFRET traces showed more frequent transitions between the open or semi-open state and the closed state. Typically, only two of three states appear in a trajectory due to the slow interconversion dynamics and limited measurement time window, but a few traces showed interconversion between all three states (Fig. 3d, upper). Consistent with the free-diffusion smFRET findings, the immobilized FRET efficiency histogram showed a significantly increased population of the closed, high-FRET conformation (42%) and attendant decreased populations

of open (26%) and semi-open (32%) conformations upon tRNA addition. The high-FRET state observed here is primarily attributable to the tRNA-bound conformation of the T-box but can also include closed apo T-boxes since they have similar high FRET values. In the presence of tRNA, the transition density plot (Fig. 3f) highlights that there are more frequent transitions between the undocked states (low/mid $E$) and the docked state (high $E$) than between the two undocked states. The transitions between low/mid and high-FRET states likely represent the tRNA binding and dissociation events, as such transitions were rare in the free T-box. Binding and dissociation transitions occurred in a single step, and we did not detect any intermediate states within the transitions, at either 1 mM or 10 mM Mg$^{2+}$, at our time-resolution of 50 ms (Fig. 3b, d, Supplementary Figs. 4 and 5).

Next, we estimated the local RNA chain flexibility using 2D FRET efficiency-donor lifetime analysis (Fig. 3c, e, Methods)[32,33]. In such plots, if the donor-acceptor distance of a molecule is constant, the distribution would be located along the diagonal line ($\tau_D/\tau_D^0 = 1 - E$). Here, $\tau_D$ and $\tau_D^0$ are the donor lifetimes in the presence and absence of the acceptor. On the other hand, if there are rapid distance fluctuations (compared to the observation time), the distribution would appear above the diagonal line[32,33] ($\tau_D/\tau_D^0 = 1 - E + \frac{\sigma_c^2}{1-E}$). Here, $\sigma_c^2$ is the variance of the FRET efficiency of the underlying fluctuating conformations. For example, the dependence of the donor fluorescence lifetime on the FRET efficiency of a random polymer chain (Gaussian chain) is a concave down curve (gray solid curves, Fig. 3c, e). The distributions of the open and semi-open conformations of apo T-boxes are close to this line ($\sigma_c^2 = 0.07$ and $0.08$ for the open and semi-open conformations in Fig. 3c and $0.08$ and $0.07$ in Fig. 3e, respectively), indicating that T-box RNA chains are flexible. By contrast, the closed conformation of apo T-box and the tRNA-bound, docked conformer, both of high-FRET, are substantially closer to the diagonal ($\sigma_c^2 = 0.009$ and $0.004$, respectively). These findings suggest that the closed T-box state is more conformationally constrained and that domain closure induced by tRNA binding rigidifies the T-box (Fig. 3c, e, light orange clusters).

Together, the 2-color surface-immobilized smFRET analysis not only corroborates the free-diffusion smFRET findings but also reveals that the open apo T-box chains are largely flexible. tRNA binding triggers rapid, single-step transitions to the closed state and reduces its conformational flexibility.

## Three-color smFRET reveals tRNA binding to all T-box conformers with distinct kinetics

One limitation of our two-color smFRET analysis is that it does not distinguish between the closed apo T-box state and the tRNA-bound closed state, both of which exhibit high FRET. To independently monitor T-box conformational changes and tRNA arrival and departure, we employed three-color smFRET spectroscopy[32,34]. To facilitate labeling, we used the anticodon stem–loop (ASL) RNA instead of the full tRNA, which binds the T-box with comparable high affinity (14 nM vs. 24 nM) since stems I–II only contact the ASL region[10]. The ASL was labeled with CF680R, which can serve as a FRET acceptor for both AF488 and AF594. Numerous ASL-binding events were detected by an increase in the fluorescence intensity of CF680R above the background, along with a concomitant decrease in the intensities of AF488 and AF594, via energy transfer (Fig. 4a, Supplementary Fig. 6). Some binding events were not accompanied by an increase in CF680R signal, likely due to incomplete labeling of the ASL or photobleaching. The FRET efficiency plotted in Fig. 4a is the pseudo 2-color FRET efficiency defined as the fraction of the acceptor intensities (i.e., the sum of the AF594 and CF680R intensities divided by the total intensity). When all three fluorophores are present in the bound state, we found that this FRET efficiency is approximately the same as the transfer efficiency from AF488 to AF594 when the bound ASL is unlabeled. This suggests that the direct energy transfer from the donor to the second acceptor in the presence of the first acceptor is negligible. In other words, the

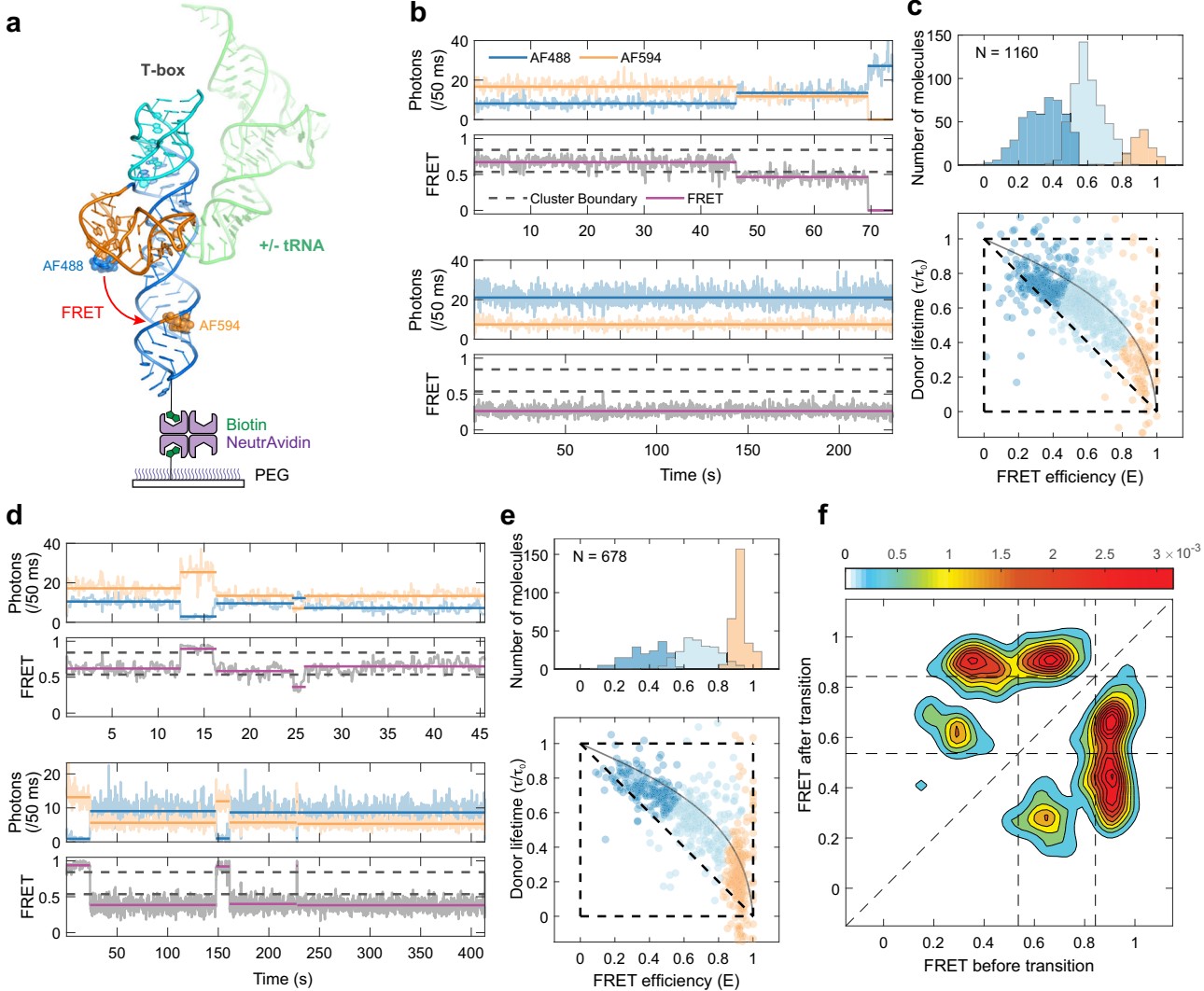

**Fig. 3 | Two-color immobilization smFRET experiment of the T-box riboswitch.**
**a** Schematic of smFRET experiment of immobilized T-box RNA in the presence or absence of tRNA. **b**, **d** Representative binned (50 ms bin time) donor (AF488, blue) and acceptor (AF 594, orange) fluorescence trajectories (upper panel) and FRET efficiency trajectories (lower panel) for immobilized T-boxes in the absence (**b**) and presence (**d**) of tRNA, respectively. **c**, **e** Upper panels: FRET efficiency histograms of three different states in the absence (**c**) and presence (**e**) of 50 nM tRNA. FRET efficiency values were corrected for background, donor leak into the acceptor channel, γ-factor, and direct acceptor excitation (see Methods). Lower panels: 2-dimensional (2D) FRET efficiency-donor lifetime plots in the absence (**c**) and presence (**e**) of 50 nM tRNA. The distributions were clustered into three states using Gaussian mixture models. The deviation from the diagonal line indicates rapid fluctuations of donor-acceptor distances (i.e., conformational flexibility) in each state. The gray solid curve shows the dependence of the lifetime on the FRET efficiency for a flexible random polymer (Gaussian chain) for comparison.
**f** Transition density plot between the three different states of WT T-box in the presence of 50 nM tRNA. Source data are provided as a Source Data file.

ASL-borne CF680R is primarily excited by AF594 but not AF488, presumably due to the much-reduced spectral overlap between AF488 and CF680R.

Using the three-color system, we monitored the T-box conformations, as measured by AF488–AF594 FRET efficiency, immediately before ASL binding events. Remarkably, we find that all three T-box conformers can directly receive the ASL, although binding events to the high-FRET, closed conformer are less common (Fig. 4a). This can stem from the fact that the closed apo T-box conformation is rare, or from potential steric hindrance of the docked state to ASL binding, or both. Interestingly, in most time traces, we find that upon ASL dissociation, the T-box molecules transitioned back into the same FRET state that was sampled prior to ASL binding. This observation indicates the presence of a memory of the T-box conformation that is not entirely lost during tRNA binding. Several trajectories showed that the same unbound state persists regardless of tRNA binding until a

single transition point between two unbound states (red arrows in Fig. 4a), which changed the unbound state FRET efficiency for the remaining trajectories.

This conformational memory can be further visualized by the transition density map (Fig. 4b). Most transitions appear on the diagonal of the plot, which correspond to events that exhibit similar FRET efficiencies of the unbound state before and after the interaction with ASL. This represents similar conformations or a memory that persists through ASL binding and dissociation. This analysis further shows that the memory effect is very pronounced at low and mid-FRET states but is much diminished or nearly absent at the high-FRET state. For binding events that start or end in the high-FRET, U3 state, the transitions are largely off-diagonal (Fig. 4b, blue-shaded region), which suggests that the FRET efficiencies or T-box conformations had changed during the ASL interaction (B1/2 to B3 or its reverse). Taken together, most T-boxes seem to "remember" their conformations before ASL binding

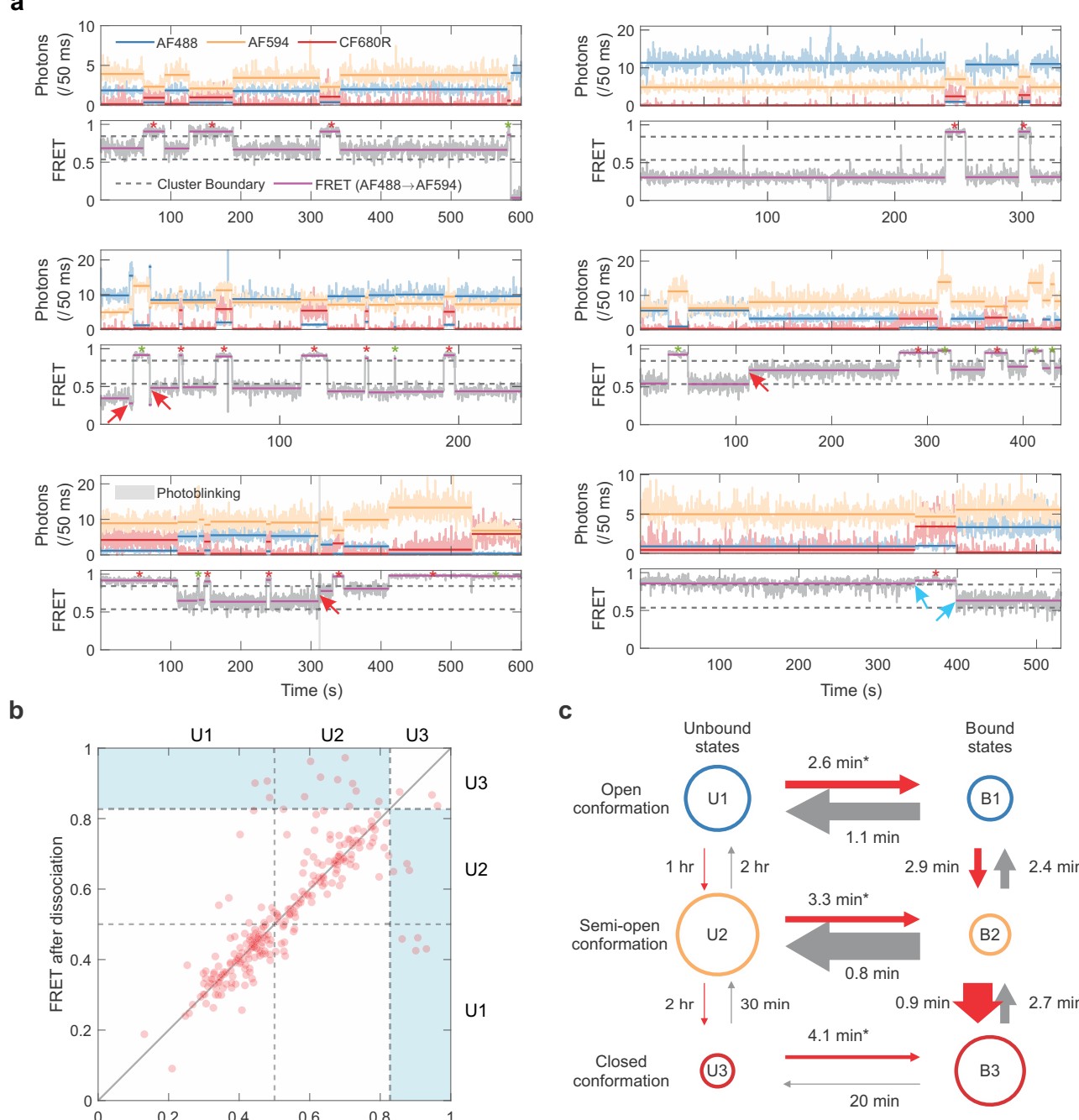

**Fig. 4 | Three-color smFRET spectroscopy of tRNA ASL binding to the T-box RNA. a** Representative binned (50 ms bin time) donor (AF488, blue), acceptor 1 (AF594, orange), and acceptor 2 (CF680R, red) fluorescence (upper) and FRET efficiency trajectories (lower). The FRET efficiency is defined as the fraction of total acceptor photons (acceptor 1 and acceptor 2), which are corrected for background, leaks, γ-factors, and direct excitation of acceptors (see Methods). Due to incomplete acceptor 2 labeling on the ASL, the bound state can appear as both 2- (binding to unlabeled ASL) and 3-color (binding to labeled ASL) segments as indicated by green and red asterisks above the FRET trajectories, respectively. After dissociation of ASL, in most cases, T-box molecules come back to the unbound state with the FRET efficiency values similar to those before binding (i.e., same unbound state) unless infrequent changes occur either in the unbound state (red arrows) or in the bound state (between the two blue arrows), indicating the presence of long-lasting memory. **b** Transition density map analysis. Correlation of the efficiency (i.e.,

fraction of total acceptor photons) values of the unbound T-box before ASL binding and after its dissociation (i.e., *E* values of two unbound states separated by a bound state). The distribution appears along the diagonal line for almost all the cases of the open (U1) state, and most cases of semi-open (U2) states, indicating the memory of the unbound conformation persists through the bound state. Conversely, the memory of the closed state (U3) is frequently lost in the bound state (transitions in the blue-shaded off-diagonal regions). **c** Kinetic model for tRNA ASL binding to the T-box RNA showing rates of interconversion (as lifetimes) between the different states (three bound states B1–B3 and three unbound states U1–U3) in the presence of 20 nM tRNA-ASL (see Supplementary Fig. 8 for the complete kinetic model accounting for incomplete labeling of acceptor 2). The area of the circle is proportional to the population of each state, and the thickness of the arrows is proportional to the rate. *binding rates with 20 nM ASL. Source data are provided as a Source Data file.

**Table 1 | Kinetic parameters of T-box–tRNA binding[a]**

| Photobleaching of acceptor 2 | Not included | Included |
|---|---|---|
| $k_{U12}$ (h$^{-1}$) | | 1.47 (0.34) |
| $k_{U23}$ (h$^{-1}$) | | 3.55 (0.87) |
| $p_{U1}$ | | 0.354 (0.013) |
| $p_{U2}$ | | 0.871 (0.011) |
| $k_{B12}$ (h$^{-1}$) | 46.8 (2.0) | 40.0 (1.7) |
| $k_{B23}$ (h$^{-1}$) | 113.4 (4.2) | 69.8 (1.7) |
| $k_{B1}$ (h$^{-1}$) | 23.4 (0.9) | 23.2 (0.9) |
| $k_{B2}$ (h$^{-1}$) | 18.0 (0.6) | 18.1 (0.4) |
| $k_{B3}$ (h$^{-1}$) | 14.5 (0.5) | 11.0 (0.3) |
| $k_{a1}$ (nM$^{-1}$h$^{-1}$)[b] | 1.17 (0.05) | 1.16 (0.05) |
| $k_{a2}$ (nM$^{-1}$h$^{-1}$)[b] | 0.900 (0.030) | 0.907 (0.022) |
| $k_{a3}$ (nM$^{-1}$h$^{-1}$)[b] | 0.725 (0.024) | 0.551 (0.014) |
| $k_{d1}$ (h$^{-1}$) | 52.6 (1.7) | 50.0 (1.4) |
| $k_{d2}$ (h$^{-1}$) | 77.4 (2.3) | 73.8 (1.6) |
| $k_{d3}$ (h$^{-1}$) | 3.08 (0.088) | 2.46 (0.052) |
| $\varphi$ | 0.807 (0.014) | 0.814 (0.016) |
| $k_{bl}$ (h$^{-1}$)[c] | | 1.83 (0.65) |

[a]Errors in the parentheses are standard deviations calculated from the curvature at the maximum of the likelihood function.
[b]$k_{ai} = k_{Bi}/[tRNA]$, where [tRNA] = 20 nM.
[c]$k_{bl}$ is the photobleaching rate of acceptor 2.

and tend to return to those conformations after ASL dissociation. This also means that the T-box-ASL complexes are not identical to each other, despite sharing the same high-FRET efficiency and overall closed conformation. Each complex individually retains a prescribed tendency to return to its pre-binding conformation upon ASL dissociation. We interpret these differences as minor, local structural differences, likely in the stems I–II junctional region, that do not interconvert during the course of binding and dissociation (Supplementary Fig. 7). Compared to the open states (U1/2) where the stems I–II junction is more structurally restrained, the closed conformation (U3/B3) has a more relaxed junction and provides more opportunities for the exchange of these local structures, and thereby loss of the memory (Fig. 4c, "Discussions").

## A parsimonious kinetic model describes dynamic interconversion between T-box states

To gain an overall dynamic picture of T-box conformational switching before and after tRNA binding, we combined the two- and three-color smFRET data to extract kinetic information employing a six-state model (Fig. 4c). This model consists of three unbound (U) states and three bound (B) states. Before tRNA binding, transitions are allowed between the open (U1) and semi-open (U2) states and between the semi-open (U2) and docked states (U3). Each unbound state is connected to one of three corresponding bound states (B1–B3). Transitions between B1 and B2 and between B2 and B3 are associated with loss of conformational memory. In the parameter optimization, we introduced 3 additional bound states to account for incomplete labeling and photobleaching of acceptor 2 (Supplementary Fig. 8, A2 dark, left column, "Methods"). The parameter optimization converged with small standard errors (<5%) except for the unbound state interconversion dynamics with a time scale of hours, which had larger standard errors of 24% (Table 1).

The resulting parameter-optimized model produces a mechanically feasible set of rates that connect the six states, explain the memory effect, account for the experimental data, and inform T-box biological function. A salient feature of the model is the loss of conformational memory due to interconversion between the bound states (Fig. 4b, c). tRNA binding to and dissociation from the open and semi-

open states of the T-box is fast, occurring over ~ 1 min, but inter-conversion rates between the bound states are also relatively fast. In particular, the rate from B2 to B3 (lifetime = 0.9 min) is comparable to the dissociation rate of B2 (0.8 min), which leads to the memory loss of B2 that appears in the upper blue region (U2 → U3) in Fig. 4b. On the other hand, the dissociation of tRNA from B3 to the apo docked state (U3) is much slower (20 min). Thus, the competing transition to another bound state (B2) is kinetically favored (2.7 min) over dissociation, driving a conformational transition to the semi-open state and loss of memory. Another prominent feature of the model is that even though tRNA binding to the docked state (U3) is the slowest, B3 is the most stable bound state among the three bound states. The rapid interconversions between the bound states, and restricted entry into U3, result in a net flux into the B3 state, which makes it the most populated bound state in the equilibrium and, consequently, the state captured by the co-crystal structure. The tRNA-bound, stems I–II-docked B3 state represents a functional intermediate whose correct folding was chaperoned by tRNA binding and is, in turn, poised to position and guide the folding of the ensuing T-box 3' discriminator domain around the tRNA 3' end to probe aminoacylation.

These kinetic characteristics translate into a physical picture consistent with prior structural and biochemical findings. Specifically, tRNA primarily binds open T-boxes, presumably by initial base-pairing interactions between the stem I Specifier and tRNA anticodon. This interaction is also suggested to occur first for Class II T-boxes that do not contain stem IIs, such as *B. subtilis glyQS*[7,27]. tRNA binding to closed T-boxes is less favored, likely due to potential steric interferences from stem II, an overabundance of concentrated negative charges near the Specifier, or both. During tRNA dissociation, the B3 to U3 transition is blocked because stem II serves as a lid that needs to move away (i.e., entering the B2 state) to allow tRNA release. This is consistent with the primary biological function of stem II, which is to reinforce stem I-tRNA interactions to prevent premature tRNA release before the 3' discriminator domain can interrogate its aminoacylation status[6,10].

We further find that the fast interconversion between the bound states provides a substantial kinetic advantage to rapidly achieve stable tRNA binding. When we simulated tRNA binding to the T-box without interconversions between the bound states, the equilibrium population of the bound states did not change (Supplementary Fig. 9). However, the time required to reach the same equilibrium increased significantly. When the system started from the three pre-equilibrated unbound states, the time required to reach 90% equilibrium binding was slowed down by 21-fold by the lack of bound state interconversion (Supplementary Fig. 9c, left). In the case that the system started solely from the open state (U1), which may approximate the state of nascent RNA during initial transcription, it was 23 times slower (Supplementary Fig. 9c, right). A brief decrease of the overall bound population was also observed, as the equilibration between unbound states decreased the U1 population and pulled back the already bound population of B1 to U1. Therefore, our modeling suggests that fast interconversion between the tRNA-bound states, in conjunction with a restricted B3 to U3 dissociation passage, efficiently channel open, tRNA-accessible T-box conformers into a closed, tRNA-retained state on the path toward the full-length T-box-tRNA complex which directs conditional downstream gene expression.

## Stem IIA/B pseudoknot orchestrates T-box conformational dynamics

Next, we sought to identify the structural origin of this unusual dynamic behavior of the T-box RNA. Stems I and II domains individually fold into stable, extended arms that are connected at their bases near the stem IIA/B pseudoknot[10]. Despite not making any tRNA contact, the deletion of the pseudoknot reduced tRNA binding by more than a 1000-fold[10]. The co-crystal structure suggests that the pseudoknot occupies a hinge-like region, contacts both stems I and II and is

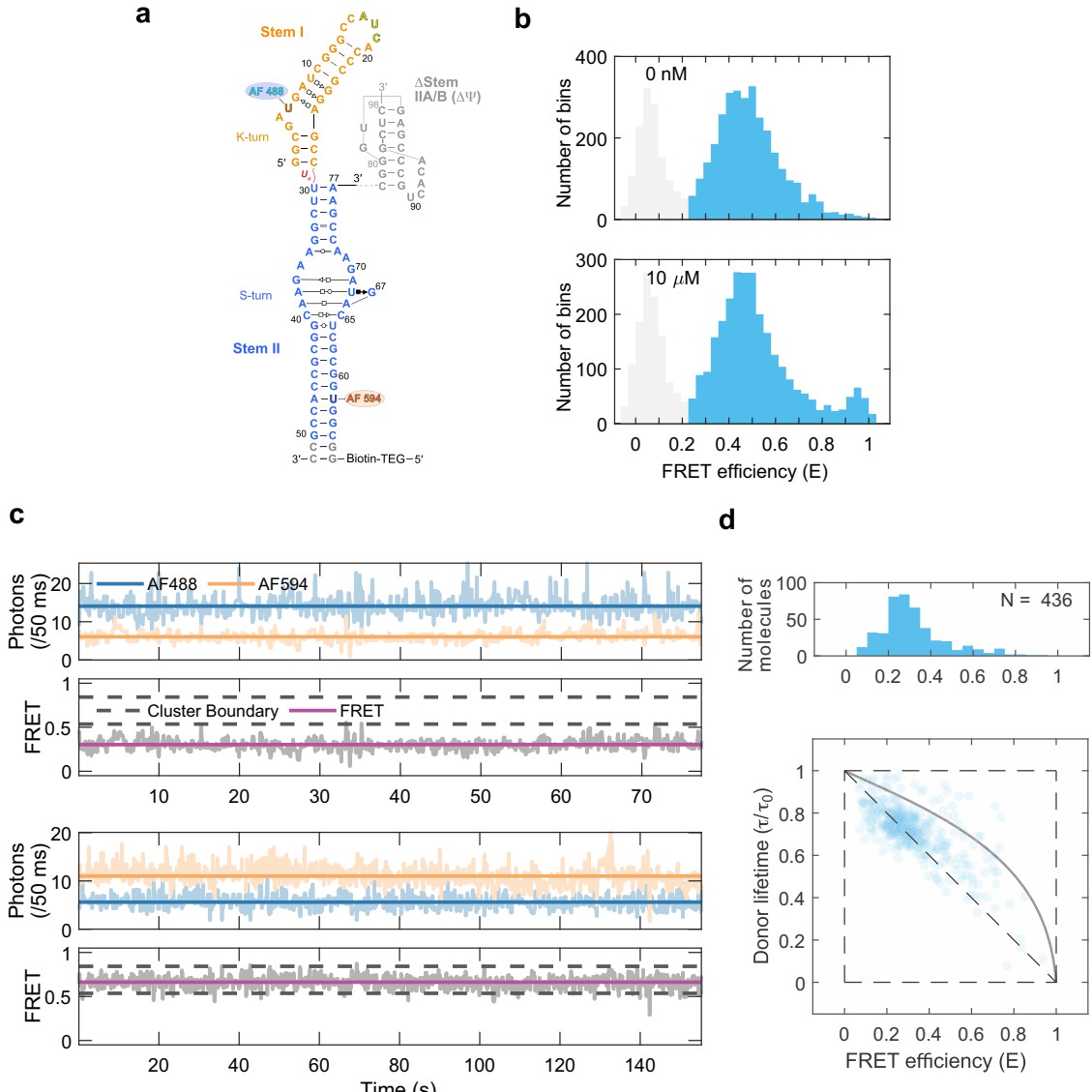

**Fig. 5 | smFRET spectroscopy of the ΔΨ T-box RNA. a** Sequence and secondary structure of the ΔΨ T-box RNA (nts 1–77, pseudoknot deletion). The location of the inserted uridine linker between stems I and II is indicated in red. **b** FRET efficiency histograms from free-diffusion experiments of ΔΨ T-box RNA in the absence (top) and presence (bottom) of 10 μM tRNA. FRET efficiency values were corrected for background. **c** Representative binned fluorescence (upper) and FRET efficiency (lower) trajectories of immobilized ΔΨ T-box RNA in the absence of tRNA. **d** Upper panel: FRET efficiency histogram of immobilized ΔΨ T-box RNA in the absence of tRNA, derived from (**c**). FRET efficiency values were corrected for background, donor leak into the acceptor channel, γ-factor, and direct acceptor excitation (see Methods). Lower panel: 2D FRET efficiency-donor lifetime plot of immobilized ΔΨ T-box RNA without tRNA. The on-diagonal distribution indicates a lack of flexibility in ΔΨ T-box. Source data are provided as a Source Data file.

well positioned to modulate their pivoting motions. Specifically, it is plausible that by stacking with stem II and projecting its L3 loop towards stem I, the mechanically robust pseudoknot can backstop stem I from swinging backward to stack with stem II and thus favor its forward motion toward stem II and increase the probability of stems I–II docking (Supplementary Fig. 7).

To test whether the stem IIA/B pseudoknot serves as a structured hinge and is responsible for the dynamic behavior observed in the apo T-box, we characterized the solution behavior of the ΔΨ T-box (1–77 nts) using free-diffusion and immobilization smFRET (Fig. 5a). First, free-diffusion smFRET revealed a narrower, down-shifted FRET efficiency distribution (Fig. 5b, upper) compared to the WT apo T-box (1–98 nts), suggesting a more homogeneous and open T-box conformation. Consistent with its previously measured, 1000-fold lower tRNA-binding affinity ($K_d$ ~ 29 μM), the addition of as much as 10 μM tRNA to the ΔΨ T-box only marginally shifted the

distribution, producing a small high-FRET peak (Fig. 5b, lower). Immobilization smFRET yielded a FRET efficiency of 0.27 (determined by fitting a Gaussian to the 90th percentile, $E < 0.54$, to discard any minor high FRET population) with a narrower distribution after correcting for donor leak and γ-factor (Fig. 5c, d). This corresponds to a donor–acceptor distance of ~64 Å, larger than the ~60 Å and 50 Å distances in the open and semi-open conformers, respectively. These observations suggest that without the pseudoknot, the T-box becomes conformationally homogeneous and assumes a conformation even more extended than the open conformation of the apo T-box. Individual time traces of the ΔΨ T-box showed constant FRET efficiencies with no observed transitions (Fig. 5c) and now occupy on-diagonal distributions in the 2D donor lifetime-FRET efficiency plot, indicating a rigid conformation ($\sigma_c^2 = 0.01$ for $E < 0.54$, Fig. 5d). Notably, despite having very low affinity and low FRET-efficiency due to unfavorable conformation,

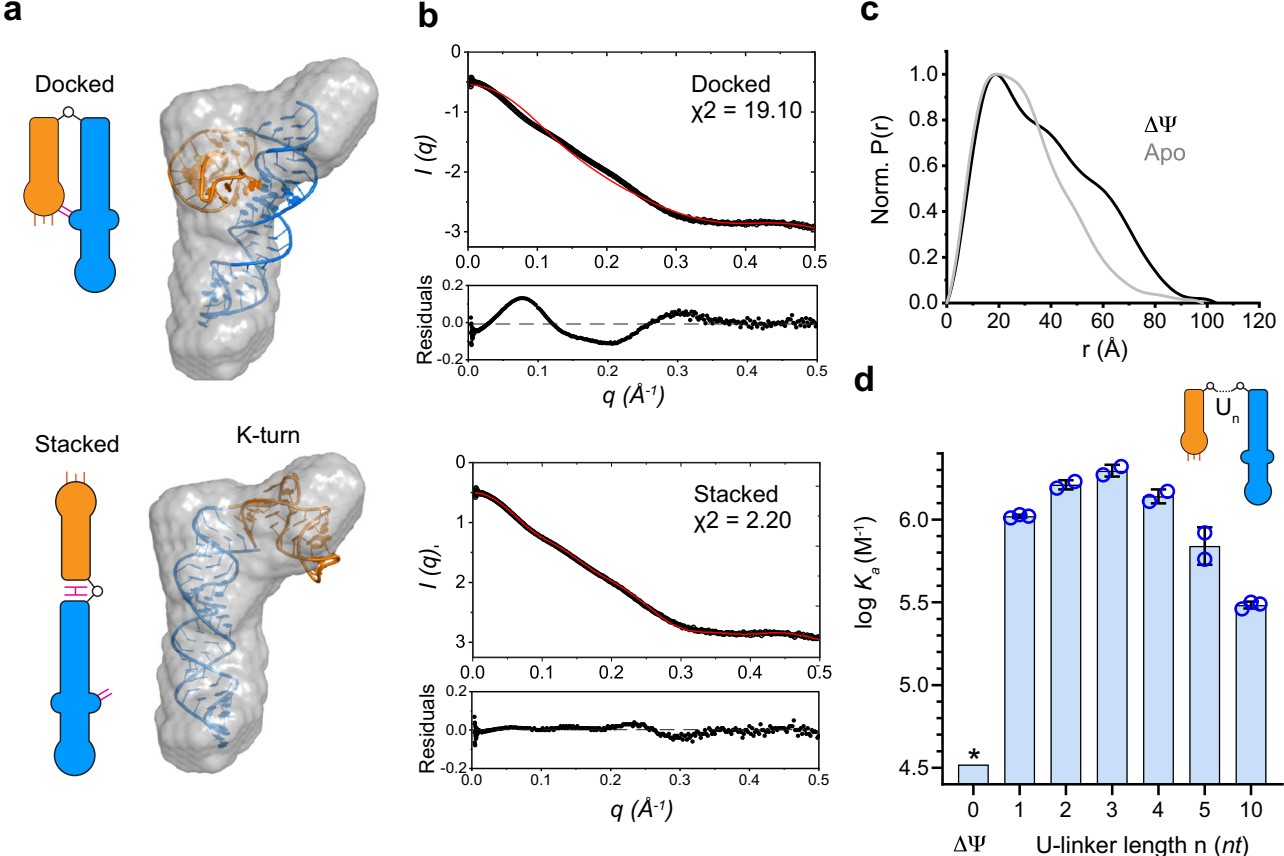

**Fig. 6 | Solution conformation of ΔΨ T-box RNA. a** Cartoon schematic (left) and structural models (right) of the ΔΨ T-box RNA rigid-body docked into the SAXS-reconstructed envelope generated by DAMMIF (gray), in stems I-II docked (upper panel) and stacked (lower panel) conformations. **b** Overlay of back-calculated SAXS scattering curves (red lines) computed using CRYSOL from the stems I-II docked structures extracted from the cocrystal structure (upper panel) or from the proposed stems I-II stacked structural model (lower panel) with experimental scattering profiles in solution (black lines). Residuals and $\chi^2$ of the fits are indicated. **c** PDDF plots for the apo T-box (gray) and ΔΨ T-box (black). **d** Effects of inserting a flexible polyuridine linker of different lengths between stems I and II of ΔΨ T-box on tRNA binding affinity. The values are mean ± S.D. from $N$ biologically independent replicates. $N = 3$ for linker lengths 1 and 10; $N = 2$ for linker lengths 2–5. * value for zero U-linker length from ref. [10] for comparison. Source data are provided as a Source Data file.

the ΔΨ T-box remains competent for tRNA binding and tRNA-induced stems I–II domain closure.

To better understand the extended, static conformation of the ΔΨ T-box, we measured its SAXS scattering profile and compared it to the stems I–II docked model extracted from the crystal structure using CRYSOL (Fig. 6a). The solution conformation was substantially different as evidenced by a high $\chi^2$ of 19.1 (Fig. 6b). Reconstruction of the molecular envelope from experimental SAXS data revealed an elongated shape of the approximate width of double-stranded RNA with a single bend, which does not agree with the docked model (Fig. 6a). The bend may be attributable to the K-turn in stem I. We reasoned that without the pseudoknot, the zero-length linker between stems I and II likely permits its end-to-end stacking, as seen in numerous multi-stem RNA structures exemplified by the HIV TAR-polyA region[35]. Thus we constructed a structural model for such a stems I–II stacked conformation, which is well constrained by the zero-length linker. Remarkably, this hypothetical model fits readily into the SAXS envelope, and its calculated scattering profile agrees with the experimental SAXS profile, producing a low $\chi^2$ of 2.2 (Fig. 6a, b). Further, the PDDF analysis shows that the ΔΨ T-box is considerably more rod-like and less compact than the apo T-box (Fig. 6c), lending further support to the stacked model. Finally, the rudimentary, unrefined stacked model predicts a donor–acceptor distance of ~71 Å, broadly in line with the 64 Å estimated from the observed FRET efficiency of 0.27 (Fig. 5d).

Together, the smFRET and SAXS data show that without the stem IIA/B pseudoknot, stems I and II are prone to stacking with each other, which prevents their docking and stable interaction with the tRNA. This model predicts that interrupting stems I–II stacking using another method other than the competing pseudoknot should also assist tRNA binding. To test this, we inserted a flexible poly-uridine linker of various lengths between stems I and II in the ΔΨ T-box and measured tRNA binding. Remarkably, all these constructs exhibited substantially improved tRNA binding by up to 59-fold ($K_d$ changed from 29 μM to 488 nM for 3-U linker, which is 20-fold away from WT T-box, Fig. 6d, Supplementary Fig. 10). This suggests that reducing stems I-II stacking by the U linkers can indeed partially substitute for the pseudoknot by facilitating their approach and docking. We further find that tRNA binding affinity initially increased with the length of the U linker, peaking at 3 nts, and subsequently declined progressively with even longer linkers. A 10-U linker yielded a $K_d$ of 3.3 μM, a mere ~9-fold improvement from the ΔΨ T-box. A few uridines appear necessary and sufficient to confer flexibility and create distance between the stem I and II termini, whereas a much longer linker incurs a progressively larger entropic penalty (Supplementary Table 1). This effect is consistent with theoretical calculations of RNA loop entropies as a function of loop lengths in pseudoknots using polymer statistical mechanics[36,37]. These findings further establish the key topological role of the stem IIA/B pseudoknot in preventing stems I–II stacking and

orchestrating a pivoting conformational ensemble poised for tRNA-induced docking and mutual stabilization.

## Discussion

The main findings of this study are: (a) the stems I–II domains present in most T-box riboswitches fold into a broad ensemble of locally flexible yet slowly interconverting conformers; (b) the incoming tRNA can bind all conformers and trigger rapid domain closure via stems I–II docking, preventing premature tRNA dissociation; (c) tRNA release is preceded and facilitated by an isomerization step into more open conformations in the bound state (from B3 to B2 and B1); (d) the stem IIA/B pseudoknot chiefly orchestrates this conformational ensemble to rapidly recruit the cognate tRNA with high affinity. Overall, the dynamic and polymorphic T-box RNA is conformationally poised to receive and subsequently trap the captured tRNA in a mechanism that resembles a Venus flytrap (Fig. 7).

Our analyses reveal three additional conformational states that the T-box stems I–II can sample, in addition to the closed, tRNA-bound conformation[10]. Since the 3' most pseudoknot is the last element to emerge from a transcribing RNA polymerase, and thus to fold[38,39], an initial stems I–II transcript without the folded pseudoknot may transiently assume the stems I–II stacked conformation inferred by SAXS and smFRET, before being outcompeted and dismantled by the stable pseudoknot. When stems I, II, and IIA/B are all present, the apo T-box primarily adopts the open and semi-open conformations and, more rarely, the closed conformation. The exact conformations of the two open conformers remain unknown but feature much larger and different separations between stems I and II. Interestingly, these conformations are relatively stable, do not frequently exchange, and tend to persist through the entire tRNA-binding and dissociation events. The source of this difference is likely the conformation of the hinge

region, including the stems I–II junction (C29–U30) and the stem IIA/B pseudoknot (Supplementary Fig. 7). Similar conformational memory effects have been observed in other structured RNAs in smFRET measurements[40]. Although their molecular origins are not well understood, they have been generally associated with complex helical junctions, as exemplified by the two-way, three-way, and four-way junctions found in the minimal hairpin ribozyme[41], cyclic-di-GMP[42], and Mn$^{2+}$ riboswitches[43], respectively.

The T-box pseudoknot presents a large interface composed of a base triple (C78•G89•U90) and a highly conserved, structured L3 loop (U90–A94, part of the "F-box" motif, Supplementary Fig. 7). U90 not only forms the interfacial base triple that stacks with stem II but also anchors stem I by a hydrogen bond between its 2'-OH and the C29–U30 backbone, at the stems I–II junction[10]. Even a conservative U90C substitution reduced tRNA binding by 20-fold, highlighting the importance of this junctional region. The rest of L3 also faces stem I, projects the flipped-out C91 nucleobase toward it, and is well positioned to backstop stem I from swinging backward to stack with stem II (Supplementary Fig. 7)[10]. Topologically speaking, stems I, II, and IIA/B form a linear array of three individual helices with flush, stably paired termini connected by zero-length linkers. Such configurations are highly favorable to engage coaxial, end-to-end stacking between the helices[11,27,44]. The central location of stem II allows the flanking stem I and stem IIA/B to compete for stacking interactions with it and, in doing so, creates bistable conformations and dynamic behavior. This stacking-centric notion is supported by the observation that a hairpin with a flush, stackable termini can effectively substitute for the pseudoknot with only sixfold reduced binding[10]. These previous findings lend further credence to the model that stem IIA/B pseudoknot serves as a topological hinge that manages the conformations of stems I and II, which expands the functional repertoire of pseudoknots[45,46].

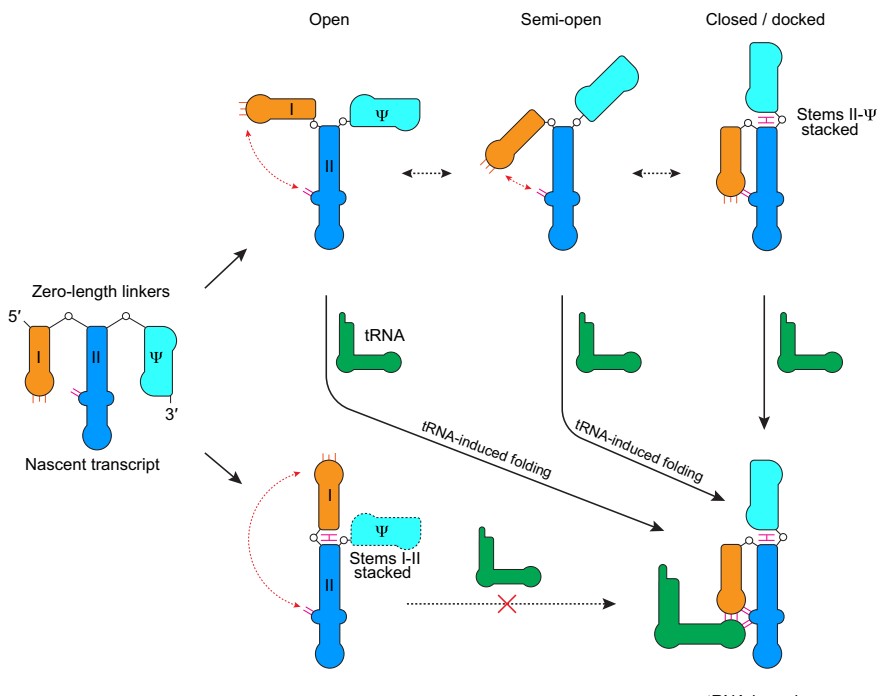

**Fig. 7 | Mechanistic model of tRNA binding to a dynamic T-box mRNA ensemble.** Most initial T-box transcripts comprise three discrete, stable helical elements: stems I (orange), II (blue), and IIA/B pseudoknot (cyan, Ψ) linked by short or zero-length linkers. Due to strong thermodynamic motivations for helical termini to coaxially stack, stem I and the pseudoknot compete for end-to-end stacking with stem II. We posit that this competition is chiefly responsible for the creation of a dynamic mRNA conformational ensemble. Among the open, semi-open, and closed

conformers, tRNA mostly engages the two open T-box conformers due to reduced steric hindrance and their larger populations. tRNA engagement triggers rapid stems I–II closure and docking, which traps the tRNA. In the absence of a stably folded pseudoknot, stems I and II stack instead, leading to a static conformation unable to retain the tRNA. Therefore, the polymorphic three-helix module of T-box mRNA is dynamically poised to rapidly recruit and stably retain its tRNA ligand, which in turn chaperones the folding of its multi-domain T-box RNA partner.

Further, we reason that subtle conformational differences in the hinge region may exert an amplified impact on the stems I–II conformations due to altered interfaces with stem I. In the open states, the stem I's base is projected to be closer to the pseudoknot, which may sterically hinder isomerization between the different hinge conformers, thus slowing interconversions. In the tRNA-bound, docked state, the base of the stem I shift away from the hinge, which potentially facilitates the conformational isomerization of the hinge, which leads to relaxation and erasure of the conformational memory. Clarifying the precise structural underpinnings of the heterogeneous, relatively stable open states awaits further structural investigations.

Structural dynamics are an intrinsic characteristic of RNA-based regulation. Alternative base-pairing between three adjacent single strands generates bistable, conditional switches for attenuators[47] and riboswitches[48]. By contrast, the widely adopted stems I–II–IIA/B element in T-boxes may provide another route to produce structural bistability, wherein alternative stacking between three adjacent helical elements generates a dynamic ensemble (Fig. 7). Compared to the first mechanism, which involves the exchange of single strands vulnerable to RNases, in-line cleavage and chemical degradation, the latter, double-stranded RNA-mediated switching does not expose single-stranded regions and maintains the integrity of the individual helices. In addition, the first mechanism generally requires energy-intensive melting and reannealing of helices under physiological conditions, which limits the length and thermostability of the helices. The latter mechanism, on the other hand, only involves the helical termini, does not require the melting of secondary structure, and thus can operate with very long helices and stable elements such as pseudoknots. Consistent with this, phylogenetic analyses of T-boxes show large variations in the length and architecture of the stem II domain[4,5,49]. We suggest that alternative or conditional helical stacking may be a general mechanism that drives global conformational changes and dynamics in structured RNAs. Additional examples include the uncharged tRNA-T-box antiterminator stacking interaction responsible for conditional gene expression[11], HIV-1 TAR−polyA helical stacking, which sequesters the 5′ cap and facilitates viral RNA genome dimerization and packaging[35,50], and a ligand-mediated helical stacking rearrangement in an FMN riboswitch[51]. This insight may guide the design of artificial RNA devices composed of multiple stackable helices (including pseudoknots) that function as effective switches or functional platforms that are both conformationally dynamic and stable against nucleases and environmental insults.

## Methods

### Labeling and purification of T-box RNAs for smFRET

For the design of the two-piece T-box RNA smFRET construct, we deleted the non-conserved distal loop of stem II and extended the stem by two G−C base pairs to increase the hybridization efficiency of the two oligos and facilitate the folding of the RNA. In this construct, the two less conserved cytosines (C6 and C58) in the T-box wild-type sequence were mutated to uridines for internal labeling with fluorophores. The 52-nt 5′ oligo and 46-nt 3′-nucleotides were chemically synthesized by Dharmacon (Horizon Discovery) Inc. with 5-aminohexylacrylamino-uridine (5-LC-NU) internal modifications containing free amine group at nucleotide positions 6 (stem I) and 58 (stem II) of the T-box. The 5′-oligo is internally labeled with Alexa Fluor 488 (donor) fluorophore at the U6 position in the K-turn bulge, which is flipped out and exposed to solvent in the crystal structure. The 3′-oligo bears a biotin-triethyleneglycol (TEG) group at its 5′-end for immobilization onto NeutrAvidin-coated glass surface and is internally labeled with Alexa Fluor 594 (acceptor) at position U58. 100 μg of the Alexa Fluor 488 or 594 dye-NHS ester solid (Molecular Probes) was dissolved in 30 μl of DMSO and used to label 4 nmol of each RNA oligonucleotide in a total reaction volume of 50 μl in 0.1 M sodium bicarbonate buffer, pH 8.6. The reactions were incubated at 37 C in the dark for 4 h, and excess unreacted dye was removed by ethanol precipitation. The 5′ and 3′ RNA oligonucleotides were annealed, and the free, unhybridized oligos were removed by gel filtration on a Superdex 200 Increase column (GE Healthcare).

### Single-molecule fluorescence experiments

Single-molecule FRET experiments were carried out using a confocal microscope system (MicroTime 200, PicoQuant) with an oil-immersion objective (UPLSAPO, NA 1.4, ×100, Olympus), a beamsplitter (z488/594rpc, Chroma Technology), and a 75 μm pinhole. Alexa 488 was excited by a 485 nm diode laser (LDH-D-C-485, PicoQuant) either in the pulsed (20 MHz) or continuous wave (CW) mode. The fluorescence signal was split into two (two-color FRET) or three (three-color FRET) photon counting avalanche photodiodes (SPCM-AQR-16, PerkinElmer Optoelectronics) using dichroic beamsplitters, 585DCXR (two-color) or 585DCXR and 670DCXR (three-color) (Chroma Technology) and through bandpass filters, ET525/50 m (Alexa 488), E600LP (Alexa 594, two-color), ET645/75 m (Alexa 594, three-color), and ET705/72 m (CF680R) (Chroma Technology). Single-molecule data were collected using SymPhoTime v5 and analyzed using Matlab 2019b.

In the free-diffusion experiments, molecules are not immobilized but freely diffuse and emit bursts of fluorescence photons when they pass through the laser focus. Samples were prepared in 1× PBS, pH 7.5. To prevent the sticking of molecules to the glass surface, 0.01% Tween-20 was added to the solution. To reduce photoblinking and photobleaching, 100 mM β-mercaptoethanol (Sigma-Aldrich), 10 mM Cysteamine (Sigma-Aldrich)[52], 2 mM cyclooctatetraene (COT, Sigma-Aldrich), 2 mM 4-nitrobenzyl alcohol (NBA, Sigma-Aldrich), and 2 mM Trolox (Sigma-Aldrich)[53] were added to the solution. Samples were illuminated in the continuous-wave (CW) mode of the laser at 30 μW. Fluorescence bursts with 30 or more photons within 2 ms bin time were considered significant bursts and analyzed.

In the immobilization experiments, biotinylated T-box mRNA molecules were immobilized on a biotin-embedded, PEG-coated glass coverslip (Bio_01; Microsurfaces Inc.) via biotin (surface)-NeutrAvidin (Thermo Scientific)-biotin (RNA) linkage. Immobilized molecules were incubated with unlabeled or CF680R-labeled tRNA anticodon stem loop (ASL) in the two- and three-color experiments, respectively. In addition to the chemicals used in the free diffusion experiments, 10 nM protocatechuate dioxygenase (PCD, P8279-25UN, Sigma) and 2.5 mM 3,4-protocatechuic acid (PCA, 37580−25G-F, Sigma)[54] were added into 1× PBS, pH 7.5 buffer to remove oxygen and minimize the photobleaching of dyes. Immobilized molecules were identified from a raster scanned image using the 485 nm diode laser at 0.2 μW, and then individual molecular trajectories were recorded at 0.02 μW until all dyes were photobleached or for a maximum duration of 10 min. To collect fluorescence lifetime information of immobilized molecules, we used pulsed mode laser excitation. However, due to the relatively slow dynamics of the T-box RNAs, we also collected trajectories in the CW mode to maximize the observation window. All experiments were performed at room temperature (21–23 °C).

### 2D FRET efficiency-lifetime analysis and clustering

For the 2D FRET efficiency-donor lifetime analysis in the two-color experiments, corrected FRET efficiencies and donor lifetimes were calculated. The FRET efficiencies were calculated by correcting the donor and acceptor photon count rates for background, leak of donor photons into the acceptor channel, $\gamma$-factor (ratio of the quantum yields and detection efficiencies of the donor and acceptor), and acceptor direct excitation at 485 nm. The donor lifetime was corrected for background. The detailed correction procedures have been described[32,55].

We clustered 2D FRET efficiency-donor lifetime distributions using Gaussian mixture models[56]. Regardless of the number of

clusters tested (up to six clusters), the low- and mid-FRET components appeared as single Gaussian clusters in the data both with and without tRNA, whereas the high-FRET component was divided into multiple clusters. This results from the very broad distribution caused by large errors in the determination of the donor lifetime due to the low donor count rate at high FRET efficiency. Therefore, we clustered the high-FRET population into a single Gaussian cluster (i.e., overall three clusters). We defined clustering boundaries as the FRET efficiency values that minimize the overlap between neighboring clusters, which were 0.45 and 0.81 for the data without tRNA, and 0.54 and 0.84 for the data with tRNA). The boundary values from the data with tRNA were used for the state assignment of each trajectory segment.

## Kinetics analysis

The clustering analysis result shows that there are three unbound states, U1 (open), U2 (semi-open), and U3 (closed). To account for the observed memory effect, we introduced three bound states with active acceptor 2 (A2) (i.e., three colors), B1, B2, and B3, which are connected to U1–U3, respectively. In addition to these 6 states, due to imperfect labeling or photobleaching of acceptor 2, there are three additional bound states without active A2 (i.e., two colors), which we denote B1*–B3*, corresponding to the three-color bound states B1–B3, respectively (Supplementary Fig. 8). As a result, the kinetic model has 9 physical states. We initially tested a more general model that also included the transition between U1 and U3 and the photobleaching of acceptor 2, but the rates of these transitions were very slow compared to other rates. Therefore, these transitions were excluded in the actual fitting model in Supplementary Fig. 8. The $9 \times 9$ rate matrix $\mathbf{K}$ for this model consists of $3 \times 3$ submatrices as

$$\mathbf{K} = \begin{pmatrix} \mathbf{K}_U^{int} - \mathbf{K}_B & \mathbf{K}_d & \mathbf{K}_d \\ \phi \mathbf{K}_B & \mathbf{K}_B^{int} - \mathbf{K}_d - \mathbf{K}_{bl} & 0 \\ (1-\phi)\mathbf{K}_B & \mathbf{K}_{bl} & \mathbf{K}_B^{int} - \mathbf{K}_d \end{pmatrix} \quad (1)$$

$\mathbf{K}_U^{int}$ and $\mathbf{K}_B^{int}$ are the submatrices describing internal transitions between unbound states and between bound states as

$$\mathbf{K}_U^{int} = \begin{pmatrix} -k_{U1U2} & k_{U2U1} & 0 \\ k_{U1U2} & -k_{U2U1} - k_{U2U3} & k_{U3U2} \\ 0 & k_{U2U3} & -k_{U3U2} \end{pmatrix}, \begin{pmatrix} -k_{B1B2} & k_{B2B1} & 0 \\ k_{B1B2} & -k_{B2B1} - k_{B2B3} & k_{B3B2} \\ 0 & k_{B2B3} & -k_{B3B2} \end{pmatrix} \quad (2)$$

where $k_{ij}$ is the transition rate from state $i$ to state $j$. Note that the rate matrices for the bound state with and without active A2 are the same. The rate matrices between the bound and unbound states (i.e., binding and dissociation of tRNA) are given by

$$\mathbf{K}_B = \begin{pmatrix} k_{B1} & 0 & 0 \\ 0 & k_{B2} & 0 \\ 0 & 0 & k_{B3} \end{pmatrix}, \mathbf{K}_d = \begin{pmatrix} k_{d1} & 0 & 0 \\ 0 & k_{d2} & 0 \\ 0 & 0 & k_{d3} \end{pmatrix} \quad (3)$$

where $k_{Bj}$ is the binding rate from U$j$ to B$j$ (and B$j$*), and $k_{dj}$ is the rate coefficient of dissociation from B$j$ (and B$j$*) to U$j$. $k_{Bj} = k_{aj}$[tRNA], where $k_{aj}$ is the association rate coefficient and [tRNA] is the concentration of tRNA. Note that the binding flux is partitioned into the three-color and two-color bound states by $\varphi$ and $(1-\varphi)$ in Eq. (1), where $\varphi$ is the fraction of active A2. $\mathbf{0}$ is the $3 \times 3$ zero matrix. $\mathbf{K}_{bl}$ is the $3 \times 3$ diagonal matrix with the diagonal elements, $k_{bl}$ that accounts for A2 photobleaching. As mentioned above, A2 photobleaching is ignored in the kinetic analysis (i.e., $k_{bl} = 0$), but there are small changes in the parameters when the photobleaching is included in the model. The parameters with and without A2 photobleaching are compared in Table 1.

Although there are nine physical states defined, the three A2-bright bound states (B1–B3) are not distinguishable, and the A2-dark bound states (B1*–B3*) are not distinguishable from one another and from the unbound high-FRET state (U3) in the experiments. Therefore, we have 5 observable states: low- (S1), mid- (S2), high-FRET states (S3, 2-color, classified based on the clustering result), A2 bright bound state (S4, 3-color), and a dark state (S5, donor blinking). Using these 9 physical states and 5 observable states, we determined kinetic parameters using the maximum likelihood method.

The likelihood function for a binned trajectory is

$$L_j = 1^{\mathrm{T}} \prod_{i=2}^{N} [\mathbf{F}(s_i) \exp(\mathbf{K}t_b)] \mathbf{F}(s_1) \mathbf{p}_{eq} \quad (4)$$

where $1^{\mathrm{T}}$ is the unit row vector, $N$ is the number of bins in the $j$th trajectory, $s_i$ is the FRET state of the $i$th bin (S1–S5), $t_b$ is the bin time, $\mathbf{K}$ is the rate matrix, and $\mathbf{p}_{eq}$ is the vector of equilibrium populations. The FRET state matrix $\mathbf{F}$ is given by

$$\begin{aligned} \mathbf{F}(S1) &= diag(1\,0\,0\,0\,0\,0\,0\,0\,0) \\ \mathbf{F}(S2) &= diag(0\,1\,0\,0\,0\,0\,0\,0\,0) \\ \mathbf{F}(S3) &= diag(0\,0\,1\,0\,0\,0\,1\,1\,1) \\ \mathbf{F}(S4) &= diag(0\,0\,0\,1\,1\,1\,0\,0\,0) \\ \mathbf{F}(S5) &= diag(1\,1\,1\,1\,1\,1\,1\,1\,1) \end{aligned} \quad (5)$$

where $diag(\mathbf{v})$ is a diagonal matrix with the elements of vector $\mathbf{v}$ on the main diagonal.

The parameters were determined by maximizing the sum of log-likelihood values of all trajectories calculated by the diagonalization of $\mathbf{K}$[57] In the parameter optimization, we assumed a detailed balanced condition, which results in the following fitting parameters:

$$(k_{U12}, k_{U23}, p_{U1}, p_{U2}, k_{B12}, k_{B23}, k_{B1}, k_{B2}, k_{B3}, k_{d1}, k_{d2}, k_{d3}, \phi) \quad (6)$$

The interconversion rates between the unbound states can be obtained by finding $k_{U12}$ ($=k_{U1U2} + k_{U2U1}$), $p_{U1}$ ($=k_{U2U1}/k_{U12}$), $k_{U23}$ ($=k_{U2U3} + k_{U3U2}$), and $p_{U2}$ ($=k_{U3U2}/k_{U23}$). From the binding ($k_{Bi}$) and dissociation rates ($k_{di}$), the ratios of the bound state populations can be determined. Using the interconversion rates between the bound states, $k_{B12}$ ($=k_{B1B2} + k_{B2B1}$) and $k_{B23}$ ($=k_{B2B3} + k_{B3B2}$), and the bound state population, the complete interconversion kinetics in the bound state can be obtained. For efficient parameter optimization, we first determined the unbound state interconversion kinetic parameters ($k_{U12}$, $k_{U23}$, $p_{U1}$, and $p_{U2}$) separately using the 2-color FRET experimental data in the absence of tRNA. Then, the rest of the whole 9-state model parameters were determined from the 3-color FRET experimental data.

## Small-angle X-ray scattering (SAXS)

The T-box RNAs and tRNAs were folded by snap-cooling and column-purified in a buffer consisting of 10 mM Tris-HCl, pH 7.5, 100 mM KCl, 10 mM MgCl$_2$, and 0.1 mM EDTA by gel-filtration on a Superdex 200 Increase column to remove any RNA aggregates. SAXS measurements were performed at room temperature at the beamline 12-ID-B of the Advanced Photon Source of the Argonne National Laboratory. Photon energy was 13.3 keV, and sample-to-detector distance was 2 m to achieve a $q$ range of $0.004 < q < 0.88$ Å$^{-1}$ where $q = (4\pi/\lambda)\sin\theta$, and $2\theta$ is the scattering angle. Measurements at three different sample concentrations were carried out to remove the scattering contribution due to interparticle interactions and to extrapolate the data to infinite dilution. Forty-five two-dimensional (2D) images were recorded for each buffer or sample solution using a flow cell, with an exposure time of 0.5 s, to minimize radiation damage and to yield an optimal signal-to-noise ratio. The 2D images were reduced to one-dimensional scattering profiles, and profiles from 45 acquisitions were selected and averaged

using the Matlab software package at the beamlines. No radiation damage was observed, as confirmed by the absence of systematic signal changes in sequentially collected X-ray scattering images. The 2D images were reduced to one-dimensional scattering profiles using the Matlab software package at the beamlines. Scattering profiles of the RNAs were calculated by subtracting the background buffer contribution from the sample-buffer profile. The solution X-ray scattering curves were back-calculated for the complex, apo, and $\Delta\Psi$ T-box RNA structures using CRYSOL with default parameters. Molecular envelopes were calculated using DAMMIF of the ATSAS suite[58].

## 2-aminopurine lifetime analysis

Lifetime measurements on 2-aminopurine (2AP) containing RNAs were performed at room temperature with 10–20 μM of RNA samples in 1× buffer containing 10 mM Tris-HCl, pH 7.5, 100 mM KCl, 10 mM MgCl$_2$, and 0.1 mM EDTA. The apo two-piece T-box RNAs were annealed and column-purified to remove free 2AP-containing oligos and any aggregates. The tRNA and A16 or A19 2AP-containing T-box complexes were formed by snap-cooling the two-piece T-box with 2-fold excess tRNA in the absence of MgCl$_2$, followed by the addition of 10 mM MgCl$_2$. The samples were column purified on a Superdex 200 Increase column to isolate the T-box-tRNA complexes and concentrated to 10–20 μM. Excess tRNA was added to the complexes to minimize the presence of dissociated tRNA-free T-boxes during the measurements. Lifetime measurements were performed on a FluoroMax Plus Spectrofluorometer equipped with DeltaTime TCSPC (time-correlated single photon counting) module (Horiba Scientific), and 2AP was excited with vertically polarized light using a 313 nm LED diode. Fluorescence emission was collected at 370 nm, with a slit width of 10 nm, until the peak photon count reached 10,000. A dilute solution of Ludox in water was used for measuring the instrument response function (IRF). Measurements were done in three different replicates, and the values reported are mean ± S.D. from the replicates. The lifetime decay curves were fit to a sum of 2 or 3 exponential functions until a good fit, as judged by $\chi^2 < 1.3$, is reached.

## Isothermal titration calorimetry (ITC)

T-box RNA and tRNA$^{\text{Ile}}$ were folded by heated to 90 °C in 1x buffer, in the absence of MgCl$_2$ for 3 min, and snap-cooled to 4 °C over 2 min, followed by the addition of 10 mM MgCl$_2$. RNA samples for ITC experiments were equilibrated in the 1× buffer by ultrafiltration 4 times. ITC experiments were performed in triplicates at 20 °C with 10–20 μM T-box RNA in the cell and 100–200 μM uncharged tRNA$^{\text{ile}}$ in the syringe, using a MicroCal iTC200 microcalorimeter (Malvern Panalytical)[7,10]. Data processing was carried out using unbiased integration software NITPIC[51] followed by SEDPHAT[52]. For binding of uncharged tRNA$^{\text{Ile}}$ by the two-piece T-box RNA, $K_{\text{d}} = 29 \pm 2$ nM, $\Delta H = -24 \pm 2$ kcal mol$^{-1}$, and $-T\Delta S = 14 \pm 2$ kcal mol$^{-1}$. These values are mean ± S.D. from two or three biologically independent samples.

## Reporting summary

Further information on research design is available in the Nature Portfolio Reporting Summary linked to this article.

## Data availability

The data supporting the findings of this study are available from the corresponding authors upon reasonable request. Source data for the figures and supplementary figures are provided as a Source Data file. PDB ID: 6UFM (https://doi.org/10.2210/pdb6UFM/pdb). Source data are provided in this paper.

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

## Acknowledgements

We thank G. Piszczek and D. Wu for their support in biophysical analyses and C. Bou-Nader, I. Skeparnias, and A. Umuhire Juru for discussions. This work was supported by the Intramural Research Program of the NIH, the National Institute of Diabetes and Digestive and Kidney Diseases (NIDDK) (ZIADK075136 to J.Z., ZIADK075113 to H.S.C.), National Cancer Institute (NCI), and an NIH Deputy Director for Intramural Research (DDIR) Challenge Award to J.Z. and Y.-X.W. We acknowledge the use of the SAXS Core facility of the Center for Cancer Research, National Cancer Institute, funded under contract 75N91019D00024. This research used resources of the Advanced Photon Source, a U.S. Department of Energy (DOE) Office of Science user facility operated for the DOE Office of Science by Argonne National Laboratory under Contract No. DE-AC02-06CH11357.

## Author contributions

K.C.S., J.Y., H.S.C., and J.Z. designed the work. K.C.S. prepared all RNA samples and performed ITC and fluorescence lifetime experiments. J.Y. performed smFRET experiments, kinetic modeling, and data analysis. L.F., X.Z., and Y.-X.W. collected and processed SAXS data. K.C.S. and J.Z. analyzed SAXS data. All authors contributed to data interpretation and paper preparation.

## Funding

## Competing interests

The authors declare no competing interests.
