## [Peer Review File · Nature Communications]

Direct observation of tRNA-chaperoned folding of a dynamic mRNA ensembleREVIEWER COMMENTS

Reviewer #1 (Remarks to the Author):

The manuscript by K.C. Suddala et al. reports the elucidation of the detailed structural dynamics between the tRNA anticodon stem loop and T-box riboswitch RNA. The orthogonal and complementary methods utilized (single molecule FRET, SAXS and time-resolved fluorescence), combined with well-designed model and control RNAs, provide a blueprint for similar kinetic mechanistic analysis of other intricate, multi-domain RNA complexes.

These studies are significant in understanding how the T-box stem I, stem II and pseudoknot multi-domain are chaperoned by the T-box RNA to arrive at the most stable complex in a kinetically expedient manner. In addition, the structural and kinetic data indicating that the pseudoknot is acting as an internal competitive inhibitor of a non-functional stem I-II coaxially stacked structure has significant implications for functional roles of pseudoknots in general.

The methodology is sound and appropriately detailed, including the detailed kinetic modeling of the multiple possible conformational states.

This is an exemplary study and well-constructed manuscript with one minor caveat: Since many of the studies were done at 10 mM Mg²⁺ (and significant conclusions drawn about the role of Mg²⁺), the manuscript should also include a discussion of the physiologically relevant concentrations of free Mg²⁺ in the cell and what the implications are of the observed results requiring a concentrations much higher than this.

Reviewer #2 (Remarks to the Author):

T-box riboswitches, non-coding RNAs found in Gram-positive bacteria, bind to specific tRNAs, sense the aminoacylation status of tRNAs in cells, and regulate the expression of downstream genes for amino acid biogenesis. Most T-box consists of Stem-I, Stem-II, Pseudoknot (Ψ), and Discriminator domain. Stem-I contains the Specifier complementary to tRNA anticodon and Stem-II stabilizes the interaction between tRNA anticodon-Specifier in Stem-I. The (Ψ) coaxially stacks with Stem-II. The Discriminator domain captures tRNA 3'-end and senses the aminoacylation status of tRNAs.

In 2019, Zhang's laboratory first reported the co-crystal structure of the Class-III T-box (ileS T-box) bound to tRNA^{Ile}. In the structure, Stem-I and Stem-II form a tight groove that interacts with the tRNA anticodon stem loop. The three-way interaction among Stem-I, Stem-II, and tRNA exhibits high affinity. Thus, the Class-II T-boxes bind to tRNA^{Ile} with higher affinity, despite the absence of stem-I distal interdigitated double T-loop motif that stacks with the tRNA elbow. However, a question has remained as to how the high-affinity three-way interactions among the Stem-I, Stem-II, and tRNA are established.

In the present study, Zhang's and Chung's laboratories have attempted to answer the question and carried out various biophysical analyses including single-molecular FRET, SAXS, and time-resolved fluorescence.

Through the analyses, they found i) smFRET and SAXS analyses showed that ileS T-box adopts multiple global conformations and tRNA-induced the compact and closed conformation of the T-box. ii) Time-resolved fluorescence decay using 2-aminopurine showed that Stem-II is not stable docked with Stem-I in the apo T-box. iii) Surface-immobilized smFRET showed that three conformational states of T-box, and tRNA addition the transition from open or semi-open to the closed state of T-box and reduces the conformational flexibility. iv) Three-color smFRET (Stem-I, Stem-II, and anticodon-stem: ASL) showed that all three T-box conformers directly receive the ASL and that upon dissociation of the ASL the T-box transit back to the same conformer. v) A parsimonious kinetic model showed that dynamic interconversion of the T-box is consistent with the previous structural and biochemical results. vi) smFRET and SAXS analyses using T-box lacking pseudoknot (Ψ) showed that the Ψ acts as a hinge which prevents Stem-I- Stem-II stacking, hereby modulates a pivoting conformational ensemble poised for tRNA-induced docking and stabilization.

Altogether, they presented the tRNA-chaperoned folding of flexible, multi-domain mRNA through a Venus flytrap-like mechanism. This study not only advances our understanding of how RNA-RNA interactions are dynamically established and regulates gene expression under nutritional starvation, but also provides a new advanced concept of how RNA can guide the folding of multi-domain RNAs.

This study is experimentally well-designed and done using solid biophysical approaches. The manuscript is also well organized and presented and high-quality. This reviewer believes that this study is merit publication in Nature Communications, and has very few comments.

Comments

i) Page5 line 134

The distance between the nucleotides labeled in Stem-I and Stem II for smFRET could be mentioned in the text here (28A ?), although the distance is described on page 9.

ii)Page lines 315-319

These sentences could be rewritten/ or rephrased for clarity.

iii) Figure 3

The reader may have some difficulty in following the order of figure panels in Figure 5 (in particular, figure 5f).

Reviewer #3 (Remarks to the Author):

T-box riboswitches are ncRNA that bind to specific tRNAs and regulate transcription or translation of

genes involved in amino acid biosynthesis, and tRNA aminoacylation. The riboswitch forms direct interactions with specific tRNA to sense the aminoacylation status. The paper focuses on *N. farcinica* ileS T-box, a class III T-box. X-ray crystal structures of Class III T-box bound to tRNA has shown a unique binding mode where stems I and II of the T-box form a tight groove for interaction with the tRNA. However, how the interaction is established is unknown. Hence, the local and global conformations of apo T-box, as well as during tRNA binding and disassociation were studied. The techniques used included two and three colored smFRET, SAXS, and time-resolved 2AP fluorescence spectroscopy.

The results demonstrated the presence of three conformations of the apo T-box riboswitch: open, semi-open, and closed. Interconversion between states is rare. Binding to tRNA results in a closed conformation, with three closed conformations possible that is dependent on the apo T-box conformation to which binding occurred. A kinetic model was designed to determine possible interconversion rates amongst the three unbound and bound states. Overall, the kinetic model for the interconversion of the T-box conformations is convincing. Multiple single-molecule and in-solution techniques were used, with observations being consistent amongst the techniques. Some major concerns need to be addressed before consideration for publication.

1. A major concern is raised if replacing the adenosines in the specifier loop with 2PA affect tRNA recognition and binding?
2. During tRNA addition in the surface immobilized smFRET, the T-box converted from open or semi-open conformation to the closed conformation. It is not clear if only single-step transition from either open state to the closed state also observed at other salt concentrations.
3. The three colored smFRET illustrated the presence of memory of the T box conformation. It was mentioned that there might be three different ASL-bound states that diverge into different apo conformations. What type of local conformations of the T-box govern how the T-box returns to the same apo state as before tRNA binding? (eg: conformations that govern B2 → U2)

Point-by-point responses to reviewer comments:

Reviewer #1 (Remarks to the Author):

The manuscript by K.C. Suddala et al. reports the elucidation of the detailed structural dynamics between the tRNA anticodon stem loop and T-box riboswitch RNA. The orthogonal and complementary methods utilized (single molecule FRET, SAXS and time-resolved fluorescence), combined with well-designed model and control RNAs, provide a blueprint for similar kinetic mechanistic analysis of other intricate, multi-domain RNA complexes. These studies are significant in understanding how the T-box stem I, stem II and pseudoknot multi-domain are chaperoned by the T-box RNA to arrive at the most stable complex in a kinetically expedient manner. In addition, the structural and kinetic data indicating that the pseudoknot is acting as an internal competitive inhibitor of a non-functional stem I-II coaxially stacked structure has significant implications for functional roles of pseudoknots in general. The methodology is sound and appropriately detailed, including the detailed kinetic modeling of the multiple possible conformational states.

A: We thank the referee for her/his generous comments and positive assessments.

This is an exemplary study and well-constructed manuscript with one minor caveat: Since many of the studies were done at 10 mM Mg²⁺ (and significant conclusions drawn about the role of Mg²⁺), the manuscript should also include a discussion of the physiologically relevant concentrations of free Mg²⁺ in the cell and what the implications are of the observed results requiring a concentrations much higher than this.

A: We appreciate the insightful question and have revised and expanded the discussion about Mg²⁺ requirements as suggested. We added a new reference (*The cellular environment stabilizes adenine riboswitch RNA structure*, by J. Tyrrell, J. L. McGinnis, K. M. Weeks and G. J. Pielak, *Biochemistry* 2013) which compared in-cell and *in vitro* structures of the adenine riboswitch using SHAPE. The implication from our findings and this reference is that the cellular environment (crowding, osmolytes, proteins, etc.) strongly promotes RNA structure formation or folding and thus significantly reduces the Mg²⁺ requirement *in vivo*.

We now state (page 6):

“This finding suggests that tRNA binding to the ileS T-box requires higher Mg²⁺ concentrations (> 10 mM) in vitro, as previously observed for this and other T-boxes such as B. subtilis glyQS^{7,19}. By contrast, the intracellular concentrations of free Mg²⁺ is estimated to be ~1-2 mM in bacteria (Tyrrell et al). This empirically observed, elevated Mg²⁺ requirement in vitro seems to be a general characteristic of complex RNA-RNA and RNA-ligand interactions. A comparative in-cell and in vitro SHAPE (Selective 2' Hydroxyl Acylation analyzed by Primer Extension) analysis of the ligand-free adenine riboswitch revealed that its in-cell and in vitro conformations correlate poorly at 1 mM Mg²⁺ (R = 0.55) but are substantially similar at 5 mM or higher Mg²⁺ (R = 0.75) (Tyrrell et al). Therefore, the in-cell environment is uniquely able to promote the folding, structuration and interactions of RNAs and obviates an otherwise much higher Mg²⁺ requirement. This effect could be mediated by molecular crowding, osmolytes such as TMAO (Trimethylamine N-oxide), or RNA-binding proteins such as YbxF which stabilizes K-turns found in T-boxes.”

Reviewer #2 (Remarks to the Author):

T-box riboswitches, non-coding RNAs found in Gram-positive bacteria, bind to specific tRNAs, sense the

aminoacylation status of tRNAs in cells, and regulate the expression of downstream genes for amino acid biogenesis. Most T-box consists of Stem-I, Stem-II, Pseudoknot (Ψ), and Discriminator domain. Stem-I contains the Specifier complementary to tRNA anticodon and Stem-II stabilizes the interaction between tRNA anticodon-Specifier in Stem-I. The (Ψ) coaxially stacks with Stem-II. The Discriminator domain captures tRNA 3'-end and senses the aminoacylation status of tRNAs.

In 2019, Zhang's laboratory first reported the co-crystal structure of the Class-III T-box (ileS T-box) bound to tRNA^{Ile}. In the structure, Stem-I and Stem-II form a tight groove that interacts with the tRNA anticodon stem loop. The three-way interaction among Stem-I, Stem-II, and tRNA exhibits high affinity. Thus, the Class-II T-boxes bind to tRNA^{Ile} with higher affinity, despite the absence of stem-I distal interdigitated double T-loop motif that stacks with the tRNA elbow. However, a question has remained as to how the high-affinity three-way interactions among the Stem-I, Stem-II, and tRNA are established. In the present study, Zhang's and Chung's laboratories have attempted to answer the question and carried out various biophysical analyses including single-molecular FRET, SAXS, and time-resolved fluorescence.

Through the analyses, they found i) smFRET and SAXS analyses showed that ileS T-box adopts multiple global conformations and tRNA-induced the compact and closed conformation of the T-box. ii) Time-resolved fluorescence decay using 2-aminopurine showed that Stem-II is not stable docked with Stem-I in the apo T-box. iii) Surface-immobilized smFRET showed that three conformational states of T-box, and tRNA addition the transition from open or semi-open to the closed state of T-box and reduces the conformational flexibility. iv) Three-color smFRET (Stem-I, Stem-II, and anticodon-stem: ASL) showed that all three T-box conformers directly receive the ASL and that upon dissociation of the ASL the T-box transit back to the same conformer. v) A parsimonious kinetic model showed that dynamic interconversion of the T-box is consistent with the previous structural and biochemical results. vi) smFRET and SAXS analyses using T-box lacking pseudoknot (Ψ) showed that the Ψ acts as a hinge which prevents Stem-I- Stem-II stacking, hereby modulates a pivoting conformational ensemble poised for tRNA-induced docking and stabilization.

Altogether, they presented the tRNA-chaperoned folding of flexible, multi-domain mRNA through a Venus flytrap-like mechanism. This study not only advances our understanding of how RNA-RNA interactions are dynamically established and regulates gene expression under nutritional starvation, but also provides a new advanced concept of how RNA can guide the folding of multi-domain RNAs.

This study is experimentally well-designed and done using solid biophysical approaches. The manuscript is also well organized and presented and high-quality. This reviewer believes that this study is merit publication in Nature Communications, and has very few comments.

A: We thank the referee for her/his detailed analysis, insightful comments and positive assessments.

Comments

i) Page5 line 134

The distance between the nucleotides labeled in Stem-I and Stem II for smFRET could be mentioned in the text here (28A ?), although the distance is described on page 9.

A: We added this distance information as suggested, and now state (page 5):

“The high-FRET value of 0.95 corresponds to a short distance of <35 Å between the fluorophores and likely represents a compact, closed conformation as seen in the co-crystal structures where stems I and II are docked via the ribose zipper, with the distance between the labeling sites being ~28 Å (Fig. 1d)”

ii) Page lines 315-319

These sentences could be rewritten/ or rephrased for clarity.

A: As suggested we have rewritten this section on the memory effect as a new paragraph, expanded and clarified it, added a new Supplementary Figure 7 to help illustrate its possible structural underpinnings, and also refer the readers to the discussion section for more details on this. We now state (page 12):

“This conformational memory can be further visualized by the transition density map (Fig. 4b). Most transitions appear on the diagonal of the plot, which correspond to events that exhibit similar FRET efficiencies of the unbound state before and after the interaction with ASL. This represents similar conformations or a memory that persists through ASL binding and dissociation. This analysis further shows that the memory effect is very pronounced at low and mid- FRET states but is much diminished or nearly absent at the high-FRET state. For binding events that start or end in the high-FRET, U3 state, the transitions are largely off-diagonal (Fig. 4b, blue shaded region), which suggests that the FRET efficiencies or T-box conformations had changed during the ASL interaction (U1/2 to U3 or its reverse). Taken together, most T-boxes seem to “remember” their conformations before ASL binding and tend to return to those conformations after ASL dissociation. This also means that the T-box-ASL complexes are not identical to each other, despite sharing the same high-FRET efficiency and overall closed conformation. Each complex individually retains a prescribed tendency to return to its pre-binding conformations upon ASL dissociation. We interpret these differences as minor, local structural differences, likely in the Stem I-II junctional region, that don't interconvert during the course of binding and dissociation (Supplementary Figure 7). Compared to the open states (U1/2) where the Stem I-II junction is more structurally restrained, the closed conformation (U3/B3) has a more relaxed junction and provides more opportunities for the exchange of these local structures, and thereby loss of memory (Fig. 4c, see discussions).”

iii) Figure 3

The reader may have some difficulty in following the order of figure panels in Figure 5 (in particular, figure 5f).

A: As suggested we have rearranged Fig. 5 panels, moving Fig. 5f to the lower right, and have panels a-c on top and d-f at the bottom.

Reviewer #3 (Remarks to the Author):

T-box riboswitches are ncRNA that bind to specific tRNAs and regulate transcription or translation of genes involved in amino acid biosynthesis, and tRNA aminoacylation. The riboswitch forms direct interactions with specific tRNA to sense the aminoacylation status. The paper focuses on *N. farcinica* ileS T-box, a class III T-box. X-ray crystal structures of Class III T-box bound to tRNA has shown a unique binding mode where stems I and II of the T-box form a tight groove for interaction with the tRNA. However, how the interaction is established is unknown. Hence, the local and global conformations of apo T-box, as well as during tRNA binding and disassociation were studied. The techniques used included two and three colored smFRET, SAXS, and time-resolved 2AP fluorescence spectroscopy.

The results demonstrated the presence of three conformations of the apo T-box riboswitch: open, semi-open, and closed. Interconversion between states is rare. Binding to tRNA results in a closed conformation, with three closed conformations possible that is dependent on the apo T-box conformation to which binding occurred. A kinetic model was designed to determine possible interconversion rates amongst the three unbound and bound states. Overall, the kinetic model for the interconversion of the T-box conformations is convincing. Multiple single-molecule and in-solution techniques were used, with observations being consistent amongst the techniques. Some major concerns need to be addressed before

consideration for publication.

A: We thank the referee for her/his detailed analysis, insightful comments and specific suggestions for improvements.

1. A major concern is raised if replacing the adenosines in the specifier loop with 2PA affect tRNA recognition and binding?

A: We thank the reviewer for pointing out this caveat. Indeed, the 2AP substitutions slightly reduced the binding affinity to tRNAs, which is expected due to minor alterations of the native base-pairing hydrogen bond patterns. We have now noted this caveat in the text, as below. However, the 2AP data is only used to compare the two apo states (not the bound state which is affected by 2AP) between where there is only Stem I *versus* where there are both Stems I and II, to further corroborate the lack of Stem-II docking in the apo state, which was primarily evidenced by the smFRET and SAXS findings. The 2AP nucleobase substitutions are not expected to directly affect Stem I-II docking, which is mediated by the backbone riboses via the ribose zipper and does not involve the nucleobases of A16 nor A19, the sites of 2AP substitution. Therefore, we believe the intended conclusion still holds that the Stems I and II are largely not docked in the apo state, despite the imperfect 2AP experiment due to the undesired but hard-to-avoid side effects of the fluorescent nucleobase analogue substitution. Compared to other commonly used fluorophores, 2AP is minimally invasive but still can exhibit minor perturbations when the interface is highly sensitive. We now state (page 8):

“The 2AP substitutions slightly reduced tRNA binding affinity as they alter the T-box base-pairing interface with the tRNA. However, neither 2AP substitution is expected to affect Stem I-II docking, which is mediated by the ribose zipper and does not involve the nucleobases of either A16 or A19.”

2. During tRNA addition in the surface immobilized smFRET, the T-box converted from open or semi-open conformation to the closed conformation. It is not clear if only single-step transition from either open state to the closed state also observed at other salt concentrations.

A: At our time-resolution of 50 ms, we observed only single-step transitions in 10 mM Mg^{2+} . To address the reviewer’s question directly, we have now also obtained and included trajectories collected at 1 mM Mg^{2+} . tRNA binding is quite rare at this Mg^{2+} concentration (please see response to reviewer #1 on Mg^{2+}). However, we are able to observe single-step transitions of the free T-box. Altogether, no intermediate state was identifiable under all conditions tested in this study. We have included this new 1 mM Mg^{2+} data in the new Supplementary Figure 5 (right), and now state in the text:

“Binding and dissociation transitions occurred in a single step and we did not detect any intermediate states within the transitions, at either 1 mM or 10 mM Mg^{2+} , at our time-resolution of 50 ms (Fig. 3b, d, Supplementary Figs. 4 & 5).”

3. The three colored smFRET illustrated the presence of memory of the T box conformation. It was mentioned that there might be three different ASL-bound states that diverge into different apo conformations. What type of local conformations of the T-box govern how the T-box returns to the same apo state as before tRNA binding? (eg: conformations that govern B2 → U2)

A: The reviewer raised a valid and probing question. However, we can only speculate on the exact nature of the local conformations, since high-resolution structural data is unavailable for any open T-box conformers. The three distinct ASL-bound states were initially identified by immobilized smFRET and subsequently supported by kinetic modeling. The observation that these three states tend to return to their respective initial conformational states suggest an intrinsic heterogeneity that is not manifest in the shared global conformation. Thus, it stands to reason that they differ in local conformations. Based on the co-crystal structure, the stem I-II-pseudoknot junction/hinge region is most likely responsible, as this region must experience local structural deformation and rearrangements when the T-box opens up, pitting the base of stem I against the pseudoknot. To better clarify this idea, we have now added a new Supplementary Figure 7 (below), where we graphically illustrate the intricate junctional structure (magenta residues) and interactions (yellow dashes) which may experience steric conflicts, deformation and create local structural heterogeneity when the T-box opens up.

We further state in the Supplementary Figure 7 legend:

“Key residues located at the Stem I-II-Ψ junction (magenta) as observed in the co-crystal structure (PDB ID: 6UFM). Hydrogen bonds are shown as yellow dashes. In the closed form of the T-box (U3, B3 states, bottom cartoon), the junction appears relaxed and free of steric clashes (green arcs). When the T-box opens up (middle and upper cartoons), the base of stem I is projected to be closer to the pseudoknot, especially the L3 loop region (U90-A94). This juxtaposition and proximity may alter or restrain the local conformation of the junction (red arcs), thereby slowing the interconversion between different conformational states of the junction. This local structural difference at the junction may persist through tRNA binding episodes and produce the observed conformational memory effect.”

REVIEWERS' COMMENTS

Reviewer #3 (Remarks to the Author):

The authors addressed most of the points raised by all reviewers.